# Learning the rules of collective cell migration using deep attention networks

**Julienne LaChance**[1], **Kevin Suh**[2], **Jens Clausen**[1], **Daniel J. Cohen**[1,2]*

**1** Department of Mechanical and Aerospace Engineering, Princeton University, Princeton, New Jersey, United States of America, **2** Department of Chemical and Biological Engineering, Princeton University, Princeton, New Jersey, United States of America

* danielcohen@princeton.edu

**Data Availability Statement:** All code used for pre-processing data, training/validating/testing the model, and post-processing for plot and figure generation can be found on GitHub at: https://github.com/CohenLabPrinceton/Attention_

## Abstract

Collective, coordinated cellular motions underpin key processes in all multicellular organisms, yet it has been difficult to simultaneously express the 'rules' behind these motions in clear, interpretable forms that effectively capture high-dimensional cell-cell interaction dynamics in a manner that is intuitive to the researcher. Here we apply deep attention networks to analyze several canonical living tissues systems and present the underlying collective migration rules for each tissue type using only cell migration trajectory data. We use these networks to learn the behaviors of key tissue types with distinct collective behaviors—epithelial, endothelial, and metastatic breast cancer cells—and show how the results complement traditional biophysical approaches. In particular, we present attention maps indicating the relative influence of neighboring cells to the learned turning decisions of a 'focal cell'–the primary cell of interest in a collective setting. Colloquially, we refer to this learned relative influence as 'attention', as it serves as a proxy for the physical parameters modifying the focal cell's future motion as a function of each neighbor cell. These attention networks reveal distinct patterns of influence and attention unique to each model tissue. Endothelial cells exhibit tightly focused attention on their immediate forward-most neighbors, while cells in more expansile epithelial tissues are more broadly influenced by neighbors in a relatively large forward sector. Attention maps of ensembles of more mesenchymal, metastatic cells reveal completely symmetric attention patterns, indicating the lack of any particular coordination or direction of interest. Moreover, we show how attention networks are capable of detecting and learning how these rules change based on biophysical context, such as location within the tissue and cellular crowding. That these results require only cellular trajectories and no modeling assumptions highlights the potential of attention networks for providing further biological insights into complex cellular systems.

## Author summary

Collective behaviors are crucial to the function of multicellular life, with large-scale, coordinated cell migration enabling processes spanning organ formation to coordinated skin healing. However, we lack effective tools to discover and cleanly express collective rules at

Networks Experimental data in the form of timelapse movies (TIFF files) and cell tracks (XML files) for HUVEC, MDCK (bulk and edge regions), and MDA-MB-231 cells may be found on Zenodo at: http://doi.org/10.5281/zenodo.4959169.

**Funding:** Partial funding support was provided by the National Institutes of Health through an NIGMS R35-133574-03 MIRA grant (held by D.J.C.; supporting J.M.L. and K.S.). The funders had no role in study design, data collection and analysis, decision to publish, or preparation of the manuscript.

**Competing interests:** The authors have declared that no competing interests exist.

the level of an individual cell. Here, we employ a carefully structured neural network to extract collective information directly from cell trajectory data. The network is trained on data from various systems, including canonical collective cell systems (HUVEC and MDCK cells) which display visually distinct forms of collective motion, and metastatic cancer cells (MDA-MB-231) which are highly uncoordinated. Using these trained networks, we can produce attention maps for each system, which indicate how a cell within a tissue takes in information from its surrounding neighbors, as a function of weights assigned to those neighbors. Thus for a cell type in which cells tend to follow the path of the cell in front, the attention maps will display high weights for cells spatially forward of the focal cell. We present results in terms of additional metrics, such as accuracy plots and number of interacting cells, and encourage future development of improved metrics.

## Introduction

Coordinated, collective migration is a hallmark, and enabler, of multicellular life. Spanning local clusters of migrating cells [1], large-scale supracellular migration across tissues [2,3], wound healing, and even coordinated cancer invasion [4,5], coordinated patterns of motion allow for complex behaviors to emerge. Understanding the collective behaviors that enable these processes can not only improve our fundamental biological knowledge, but can allow us to more effectively detect abnormalities and pathologies, and perhaps make better prognostic or diagnostic assessments [6,7]. To realize this potential, we need to first be able to define the underlying 'interaction rules' that give rise to something like humans queuing in line, jammed penguins clusters shuffling on the ice [8], and metastatic cancer cells disseminating through healthy tissue [7]. However, detecting and classifying these behaviors is not straightforward, as different fields rely on unique tools, analyses, and lexicons. Here, we explore the utility of translating deep attention networks, previously used to reveal rules of collective motion in tens of schooling fish [9], to thousands of interacting and migrating cells of disparate origins with unique patterns of motion—blood vessel endothelial cell sheets; kidney epithelial cell sheets; and large ensembles of metastatic breast cancer cells (representative motion trajectories are shown in Fig 1A–1C, with movies in S1–3 Movies, respectively). We follow the methodology of Heras et al. [9] in both modeling and analysis. Crucially, this technique requires only cell trajectory data rather than any assumptions of underlying models or dynamics.

As collective behaviors play out at the ensemble level, approaches from statistical mechanics are used to great effect to identify patterns in collective cell motion. For instance, early applications of measures such as velocity correlations to assess order and directionality in bird flock and fish school dynamics [10–12] have since been repurposed for collectively migrating cells [13–17]As an example, we computed the ensemble speed, velocity cross-correlations (Fig 1D–1E), and mean-squared-displacements (S1A Fig) for three radically different cell types—epithelia, endothelia, and metastatic breast cancer cells. While all three systems exhibit similar mean migration speeds, they deviate in the other metrics. MDCK epithelia and HUVEC endothelia cells are known to migrate collectively and present very similar, slowly decaying velocity cross-correlations; indicative of long range correlated motion (Fig 1E). Metatstatic MDA-MB-231 cells, by contrast, show a much more abrupt drop in correlation over distance (Fig 1E) indicating much smaller coordinated domains. Further analysis via the mean-squared-displacement (MSD) can also allow biophysical classification of collective migration strategies by categorizing motion as super-diffusive (endothelial), highly diffusive (metastatic cells), or a mix of super-diffusive and caged (epithelia) as in S1 Fig. In this vein, others have used

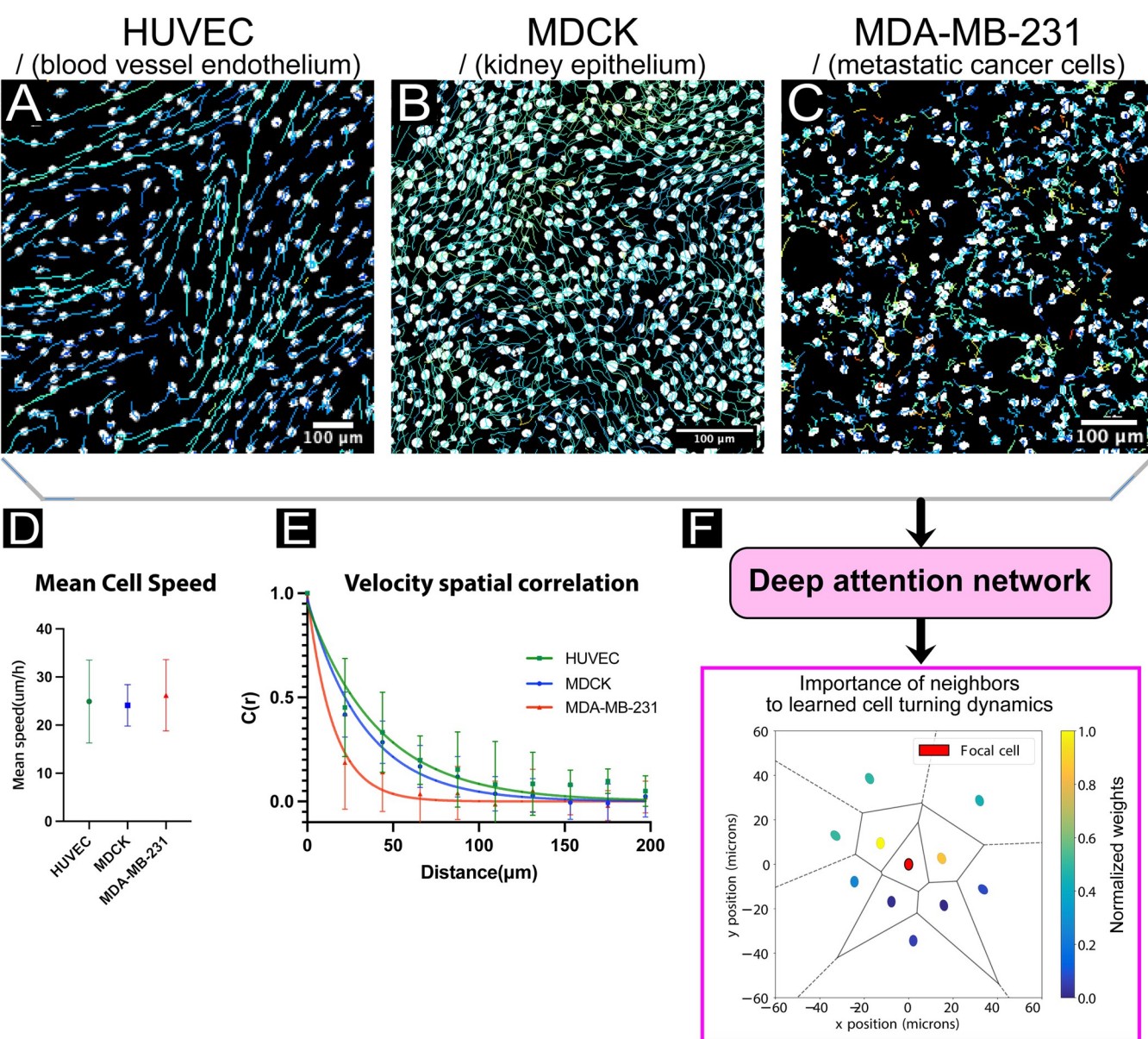

**Fig 1. Cell trajectory data reveals collective rules.** (A, B, C) Representative cell trajectories within living tissues, from human umbilical vein endothelial cells (HUVEC), Madin-Darby Canine Kidney cells (MDCK), and epithelial, metastatic human breast cancer cells (MDA-MB-231), respectively. All three cell lines exhibit visually distinct dynamics: the HUVECs tend to have strongly correlated and directed leader/follower behavior; while MDCKs exhibit more complex coordination patterns and lack the directedness of HUVECs (e.g. see [23]); and the MDA-MB-231's lack coordination with neighbors. Scale bars are 100 μm. **See S1–3 Movies.** (D) Classical collective analysis techniques reveal some group characteristics, such as mean speed or (E) velocity cross-correlations. (F) Deep attention networks trained on cell trajectory data can directly reveal new types of collective information, such as the learned relative influence of neighboring cells to forward motion of a focal cell. Here, the agents in front of the focal cell have higher weights, *W* (see Eq 1), with relative directions determined by agent trajectories. Cell position is representing using nuclei centroids and black lines indicate Voronoi cells (see *Methods*).

measures of self-diffusivity and internal deformations to describe the glass-like dynamics of such systems, quantifying the similarities between fluid-like behavior of cell sheets over long time scales and solid-like behavior at short time scales with supercooled fluids approaching a glass transition [18]. However, these are all bulk metrics describing the overall rheology or coordination of the population rather than providing data that can be interpreted at the level of the 'rules' followed by a given cell in the population.

Further, numerous classical physical models have been developed in an attempt to describe collective cell migration, including lattice, phase-field, active network, particle, and continuum models [19], with some scholars moving towards the utilization of reinforcement learning to construct agent-based models in recent years [20–22]. A hallmark of all of these approaches is that they are rooted in physical assumptions and first principles. Since the classical approaches are constrained by parameter complexity, enabling scientists to write mathematical descriptions of the system and obtain an intuitive grasp of the model components, they are often unable to effectively or efficiently capture high-dimensional interaction relationships.

Deep learning, in contrast to physics-based approaches, offers intriguing potential for the automated discovery of collective behaviors based solely on relatively simple biological input data, such as cell migration trajectories. This approach can reduce researcher bias and the need for formalized models and, when paired with interpretable data output and visualizations, can express clear patterns of behavior in complex systems. Thanks to recent advances in high-throughput, high-content microscopy [24,25] and image processing [26–30], rich visual features can be extracted from massive, dynamic populations of cells, providing a wealth of the kind of raw data through which deep learning approaches excel at sifting. Unfortunately, while deep learning methods can be structured to capture high-dimensional functions, they are often difficult to interpret. To address this, recent efforts have employed a newer approach— deep attention networks [31–33]—to reveal collective rules in schools of zebrafish (*Danio rerio*). Critically, such attention networks can be structured such that system dynamics can be learned using a function which is parameter-rich while still requiring only a small number of inputs and outputs [9]. In this study, we apply deep attention networks to large cellular ensembles in an attempt to identify patterns of cellular attention and underlying collective rules. Specifically we ask the following question of the deep attention network: given a 'focal' cell in a group of cells of a given type, where the 'focal' cell is simply the primary cell of interest and interacts with $n$ nearest neighbor cells, to which other cells does the focal cell seem to "pay the most attention" when deciding how to turn? More technically: which neighboring cells have greater relative influence on the forward motion of the focal cell, according to the dynamics learned by the model (Fig 1F)?

It is this *interpretability* of deep attention networks which is so crucial to the identification and classification of collective rules. For any given focal cell, asocial data ($\alpha$, trajectory data from the focal agent) and social data from $n$ nearest neighbors in the collective ($\sigma_i$, relative positions, velocities, accelerations of neighbors) are integrated by the deep attention network to predict the future motion of the focal cell—whether it will turn left/right, for example. Here, interpretability is gained because the network is structured in the form of an equation which combines a pairwise interaction function, $\Pi$, with a standard weighting function, $W$, as follows:

$$z = \sum_{i=1}^{n} \Pi(\alpha, \sigma_i) \frac{W(\alpha, \sigma_i)}{\Sigma_j W(\alpha, \sigma_j)}, \qquad (1)$$

where $z$ is a *logit*, a single value indicating a left or right turn of the focal agent after a fixed prediction timestep, and $n$ is the total number of nearest neighbors [9]. The logit differentiates between forward motion in the left hemisphere with respect to the focal agent's forward heading, and forward motion in the right hemisphere. Since the pairwise interaction, $\Pi$, and weight function, $W$, may vary according to the social and asocial variable inputs, various collective interaction rules may be recovered by observing how these functions and the output logit $z$ change as the inputs vary: see analyses of simulated and experimental swarm systems in [9]. These analyses may be further supplemented or validated using classical techniques, such as

assessment of mean speeds, velocity cross-correlation and MSD within a migrating collective (Figs 1D, 1E and S1A). For cellular systems, we focused on *attention maps*, which represent the output of the weight function, $W$, for many nearest neighbors, thereby allowing us to determine for any given cell which neighbors most strongly influence the future motion of the focal cell according to the trained deep attention network model (Figs 1F and S2). Combining these maps over many focal cells provides a sense of the ensemble migration rules.

To first build confidence in this approach from complex collective migration systems, we tested network performance against the classic Vicsek agent-based model of collective motility. Here, agents move with constant speed and adjust their heading to the average of all other agents within their perception zone, typically a circle of a given radius, and we implemented this in a manner that allowed us to directly pass trajectory data of individual agents to the attention network (see Methods for our simulation parameters and approach). First, we confirmed that the network could recover the largely radially symmetry attention zone of the classic Vicsek model (S3A Fig). Next, and more striking, we implemented specific narrowed perceptual zones, reducing any given focal agent's awareness to a small sector of different widths and directions. To a human observer, this subtle shifts in perceptual zone are impossible to detect by observation alone, and would be quite difficult to extract using classical methods. However, the network was able to accurately recover each unique perceptual zone we tested (S3B–S3D Fig). Together with boids model simulation results in Heras et al. [9], these data validate the efficacy of attention networks and allowed us to move forward with cellular analyses.

## Defining and constraining the problem: cellular model systems selection

To determine if deep attention networks reveal useful information from cellular systems, we selected three standard tissue models commonly used as gold standards in collective cell behavior studies. First, we considered sheets of cultured Human Umbilical Vein Endothelial Cells (HUVECs) whose hallmark is the development of strongly aligned 'trains' of cells migrating in a leader-follower fashion with weak lateral interactions. Next, we compare these to kidney epithelial sheets (MDCK cells)—one of the most well-studied living collective systems whose cells classically produce coordinated, swirling domains. Finally, as a negative control we attempt to extract the rules for metastatic breast cancer cells (MDA-MB-231) as metastatic cells behave more mesenchymally, or individualistically, and are known to lack key cell-cell interaction proteins [34–36]. Representative collective motion trajectories of these three cell types are shown in Fig 1A–1C, respectively.

To a human observer, these tracks are visually distinct, but relating the ensemble visual patterns to which neighbors are most influential to the future motion of a given focal cell, as a function of the learned dynamics, is not simple. Classical group-level analyses can be used to quantify and understand some of these patterns, as discussed earlier with respect to correlations and migratory dynamics (Figs 1D, 1E and S1A). However, while classical ensemble analyses are powerful and can, and should, be used to learn more about these systems, ultimately they cannot directly answer the question we posed above about how the dynamics of a given focal cell are influenced by specific nearest neighbors. To address this, we trained a deep attention network using cell trajectory data from long, time-lapse recordings. The trained network can then directly determine the number, location, and characteristics of the most important neighbors for a focal cell, as shown in Fig 1F where a focal agent is shown with its 10 nearest neighbors. Here, the neighbors are colored according to the (normalized) aggregation weights ($W$) from a model trained on tissues of the same type (MDCK). Due to the structure of the

network, the colors indicate the relatively higher influence of neighboring cells forward and to the sides of the focal cell for influencing migration behaviors (representative snapshots from our other model systems are shown in S2 Fig). In this study, we focused on aggregating these snapshots across many focal agents- and their respective neighbors- to produce even more informative attention maps.

Our approach here was to examine and compare attention maps for different cell types and analysis conditions in order to determine the feasibility of using deep attention networks for collective cell behavior insights, and to provide design guidelines for optimal parameters for this application. From the network perspective, we investigated prediction time intervals, image sampling frame rates, number of neighbors accounted for by the network structure, and blinding to certain input parameters; in each case using archetypal cell types for validation. Having validated the network, we then explored within a single model system how tissue age and where a cell is located within a tissue of a given shape affected neighbor interactions rules. When possible, we compare our findings from the network-produced attention regimes to results from classical analytical methods. Overall, our results demonstrate that deep attention networks offer a powerful, complementary approach to classical methods for analyzing cellular group dynamics that can reveal unique aspects of how specific cell types interact at the tissue level.

## Results

### Demonstration of attention maps for canonical cell types

To validate the deep attention networks on canonical experimental model systems, we first compared network performance on HUVEC endothelial sheets and MDCK epithelial sheets. Representative fluorescence images of each cell type are shown in Fig 2A highlighting VE- or E-cadherin at cell-cell junctions. This context is important to understand that highly collective cells tend to be physically coupled to each other through mechano-sensitive junctional proteins [37]. To standardize all model systems and analyses and provide sufficient replicates, we grew tissues in microfabricated circular stencil arrays and seeded a sufficient number of cells to reach confluence before analysis. Specifically, we incubated cells within these stencils for ~16 hrs to ensure formation of confluent tissues with no gaps (all cells should have contiguous neighbors), and then removed the stencils to allow the tissues to grow out. This approach is well characterized for these cell types and collective cell behavior studies [15,38] and generates tissues with distinct boundary and bulk regions. We then performed automated, phase-contrast time-lapse imaging over 12–24 hrs. Nuclei were segmented using a convolutional neural network [39] (MDCK), or live nuclear imaging (HUVEC, MDA-MB-231), and then tracked to generate trajectories for every cell over the course of the experiment, after which the data were ready for attention analysis.

Raw trajectory data were processed to determine the social and asocial variables as input to the attention network, as well as output turning logits. Data were split into training, validation, and test sets, and all results provided are reflective of the test set (with the exception of training loss and accuracy plots in S4 Fig). Raw data, code as adapted from Francisco J. H. Heras et al. [9], and documentation are provided at GitHub and Zenodo (see *Methods*). To best visually capture an attention map for a given tissue type, we integrated the individual attention snapshots (e.g. Fig 1F) over 10,000 individual cells from across the different replicates and interpolated the attention weights in space ($x,y$ position of neighboring cells of the focal cell) as a contour plot as shown in Fig 2C–2C'. For our initial analyses, the attention networks were structured to analyze only the 10 nearest neighbors of a given focal cell, trajectories were sampled every ten minutes, and the prediction interval was 20 minutes. The importance of these parameters and related design considerations will be discussed in the following sections.

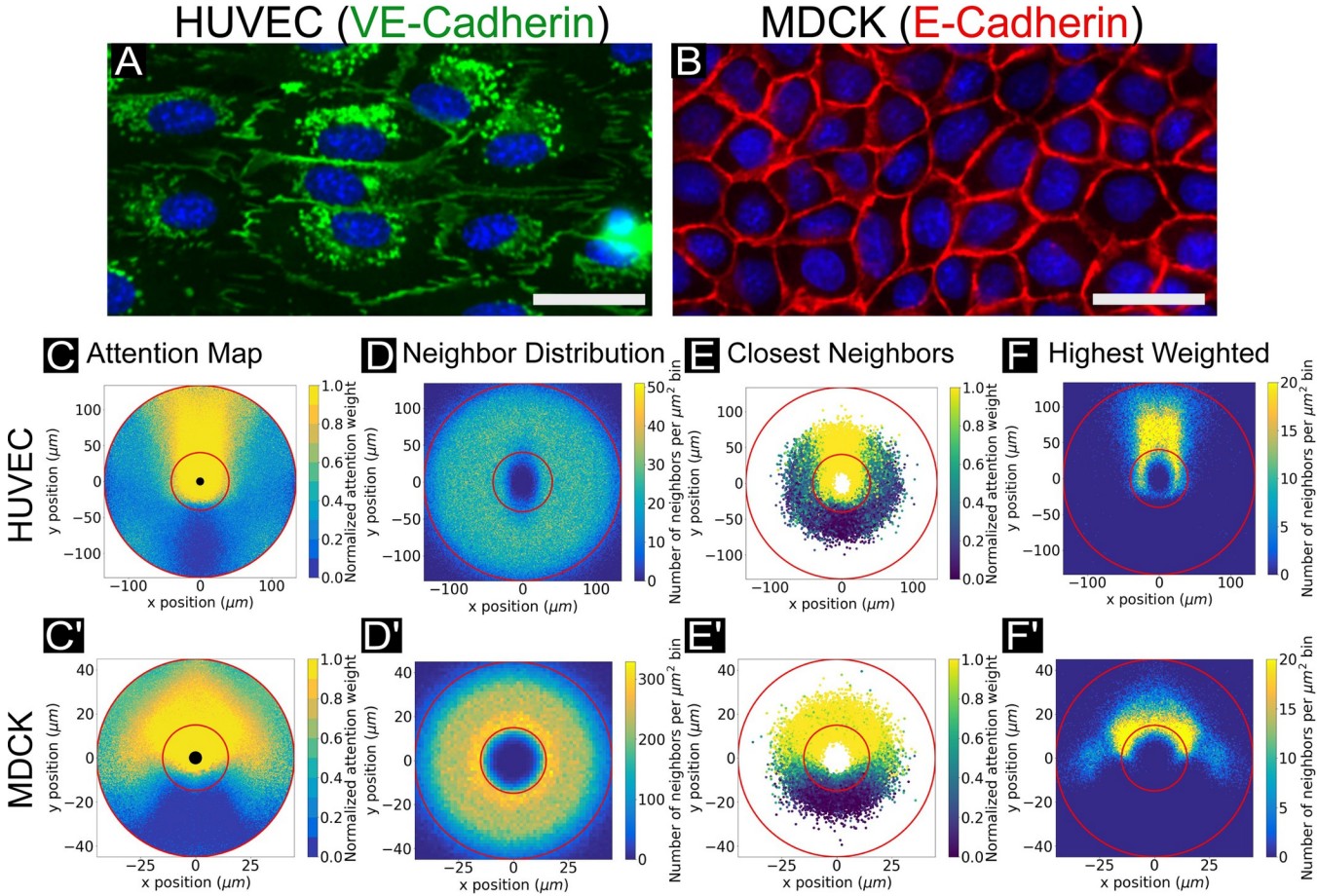

**Fig 2. Network attention across canonical cell types.** (A) VE-cadherin cell-cell junctions are indicated in red, with cell nuclei indicated in blue. VE-cadherin fingers in HUVEC cells indicate the direction of coupling between leader and follower cells. Scale bar is 30 μm. (B) E-cadherin cell-cell junctions are indicated in red, with cell nuclei indicated in blue. E-cadherin walls do not visibly indicate coordination as VE-cadherin in HUVECs. Scale bar is 30 μm. (C, C') Representative attention weight contour plots are shown for HUVEC (*top*) and MDCK cells (*bottom*). For all conditions, normalized weight maps are shown. The HUVEC attention map highlights the tendency of HUVECs to "follow the leader", with high attention weight values assigned to cells directly in front of the focal cell, spatially. By contrast, the MDCK map displays higher attention weights forward and to the sides. Central black circles indicate the radius of the closest neighbor location in the dataset. For all plots shown, networks were structured to encompass 10 neighbors, with trajectory timesteps of 10 minutes and forward prediction times of 20 minutes. (D, D') Histograms showing the distribution of data points (neighbor cell locations) from which the attention maps in (C, C') were generated. In (C, C') and D, D'), thin red circular lines indicate the annulus in which the bulk of the data (5%-95%) lies by radius (see *Methods*). Network results are expected to be more reliable within this region. Histogram bins span 1 μm$^2$. (E, E') Scatter plots showing locations of the closest neighbor to a focal agent across all focal cells, colored by normalized attention weight. (F, F') Histograms showing the locations of only the neighbor with highest weight value for each individual focal cell. Histogram bins span 1 μm$^2$.

Looking first at the attention maps for HUVECs and MDCKs immediately revealed clear differences in collective attention between the two cells. Starting with HUVECs, the network determined the most influential neighbors to be overwhelmingly directly ahead of a given focal cell (Fig 2C) with very little influence from either side or the rearward neighbors. An advantage to working with HUVECS is that there is a clear biological basis for such behavior— polarized fingers of VE-cadherin (visible in Fig 2A) protrude from the leading edge and into the trailing edge of any given cell in a train. Such fingers are not observed at lateral edges, resulting in the highly directed 'trains' of cell migration so characteristic of HUVECs [40]. Intriguingly, the lack of rearward attention captured in the map reveals information not immediately recoverable by classical methods, which have previously indicated only that velocity correlations exist between a focal agent and both its forward and rearward nearest

neighbors, respectively [40]. Similarly, fluorescence imaging data alone was unable to reveal the relative influence of front versus rear fingers. By contrast, the network can decouple simple directionality correlations (e.g. cells are moving the same direction) from attention, revealing that the immediately forward cells *specifically* have far more influence on endothelial cells than lateral or rear cellular neighbors. By contrast, MDCK cells exhibited a far broader angle of influence (Fig 2C'), with the most influential neighbors apparently lying within a ~160˚ sector around a given focal cell. This again agrees with biological context, given that epithelial cells tend to adhere strongly to neighbors on all sides (Fig 2B) and move through arcing turns as large, correlated domains [15,16,38]. Attention maps generated after different training steps (in epochs) are shown in S5 Fig, and demonstrate convergence of the attention maps to the fully trained result; these maps correspond to the training validation accuracy plots shown in S4 Fig. With increasing accuracy, the attention maps refine to produce clearer patterns of learned relative neighbor influence by spatial location. Attention maps are additionally generated for slower and faster cells in the system independently (above/below a median speed threshold), but no structural difference in the plot was observed (see S6 Fig). The network capacity to capture specific narrowed perceptual ranges were additionally validated in simulation utilizing a Vicsek model (S3 Fig, *Methods*). To a human observer, the perceptual zones of the agents are impossible to detect from the simulation output. In conjunction with the simulation results in [9], this provides support for attention networks as a valuable tool for accurately extracting perception information encoded in trajectory data.

Attention maps are interpolated over the population and could potentially be biased if cells were irregularly distributed spatially. To rule this out, we analyzed distributions of neighbor locations (Fig 2D–2D') for the data used to calculate attention maps (Fig 2C–2C') These plots indicate where the 10 nearest neighbors of any given focal cell were likeliest to be found, bearing in mind that all analyzed populations were confluent (the cells fully tiled the 2D space). Additionally, we indicate via thin red circular lines the annular region within which the bulk of the data points (5%-95%) lie as a function of radius (Fig 2C–2F'). Supplemental analogous histograms of the closest neighbor plots for all three main cell systems are provided in S7 Fig for comparison. The trained attention network weights are expected to be more reliable within this annular region than in external regions where data points were too sparse to ensure adequate modeling. In HUVECs, these neighbors appear to be evenly distributed within ~100 µm directly ahead of the focal cell. In MDCKs, however, the neighbor distribution showed a distinct gradient, with likelihood of neighbors peaking within an ~15 µm radius of the focal cell, and then dropping off by ~50 µm. However, in both cases neighbors are evenly angularly distributed about a given focal cell, meaning that the anisotropic attention maps are not due to irregular neighbor distributions, and must instead genuinely reflect spatial patterns of cellular attention. Finally, attention maps were additionally generated for slower and faster cells in the system independently (above/below a median speed threshold), but no structural difference in the plot was observed (see S6 Fig).

Attention networks offer the flexibility to investigate both population and individual cell details, so we next raised the following question: is the closest nearest-neighbor always the most important? We addressed this by comparing the attention weights of only the single closest nearest-neighbor of each focal cell to attention maps showing the locations of only the most highly influential neighbors. Fig 2E–2E' are scatter plots of only those neighbors which are the single closest neighbor by radial distance to the focal agent, with focal agents consistent with those shown in Fig 2C–2C'. The scatter points are colored by normalized attention weight. Fig 2F–2F' are histograms indicating the location of only the single highest weighted neighbor to those same focal cells. Here, we found that while the nearest neighbors themselves were uniformly distributed around a given focal cell, the relative importance of a given

neighbor depended on both proximity and orientation, rather than proximity alone, and this trend applied to both of our archetypal tissues. When considered together, the kinds of analyses shown in Fig 2 can provide a unique, rich view of the interaction network and decision making within tissues.

## Learned important neighbors and neighborhood size

Tissues such as the epithelia and endothelia serve a barrier and structural function, meaning they must maintain integrity. To accomplish this, cells tile together to form confluent layers with no empty space [41,42]. In such tissues, the dominant signaling appears to be largely mechanical, with traction strains coupled through the substrate and cell-cell tension coupled through cell-cell adhesion proteins such as the cadherins [43,44]. In such barrier tissues, a focal cell only directly communicates with those neighbors to whom it is physically adhering, while longer range force coupling requires that mechanical information be relayed from cell to cell. Hence, confluent tissues acquire distinct packing geometries, with a key metric being the number of physically contacting nearest neighbors [45,46]. This raises an interesting question from the perspective of an attention network: what is the relative influence of contiguous neighbors versus neighbors farther afield?

We first investigated this using our MDCK epithelial model as significant biophysical data exist on cell-cell adhesion, packing structure, and force coupling. Here, we used cell nuclei to tile a tessellation, from which we calculated the total number of physically contiguous neighbors for each focal cell (*Methods*). These data are compiled in Fig 3A, showing that MDCKs typically possess 5–6 contiguous nearest neighbors. The deep attention networks, however, may be flexibly structured to take input information from arbitrarily large groups of neighboring cells in order to predict turning motions of the focal agent. Thus, the network may have direct information pertaining to cells which the true biological agent may not physically contact. It is essential to remember this key distinction as larger network structures are explored: predictive power in the model may not directly indicate causative biological influence. For all analyses shown for MDCK cells in Fig 3, the corresponding neighbor distribution, closest neighbor, and highest weighted neighbor maps are shown in S8 Fig. *For the matching study with HUVEC endothelial cells, see S9 Fig.*

By utilizing a function of the inverse of the typical weight, $w_t$, as in [9]:

$$N_{total} = \frac{1}{w_t} = e^{-\sum_i w_i \log(w_i)},\tag{2}$$

the most important neighbors (as learned by the network) to the turning dynamics may be estimated. The number of total and "important" interacting agents are shown in the histogram in Fig 3B, wherein a peak in the number of important interacting agents may be observed at 5 neighbors, indicating the bulk of influence to the learned dynamics even when the network has access to information from ten neighbors in total. These data add context to the findings in Fig 2 indicating that a combination of proximity and location determines relative influence for a given neighbor.

To assess the impact of providing trajectory information to the network from larger sets of nearest neighbors (structurally, more pairwise-interaction and aggregation subnetworks), we provide network accuracy results from networks spanning 5–50 neighbors in increments of 5 (Figs 3D and S10 for additional accuracy results) and representative attention plots from networks structured to account for 10, 20, and 30 nearest neighbors in total (Fig 3E–3E"). Additionally, we consider different prediction time intervals to explore how attention network accuracy relates to predicting turning dynamics 20 minutes vs. 60 minutes into the future. In

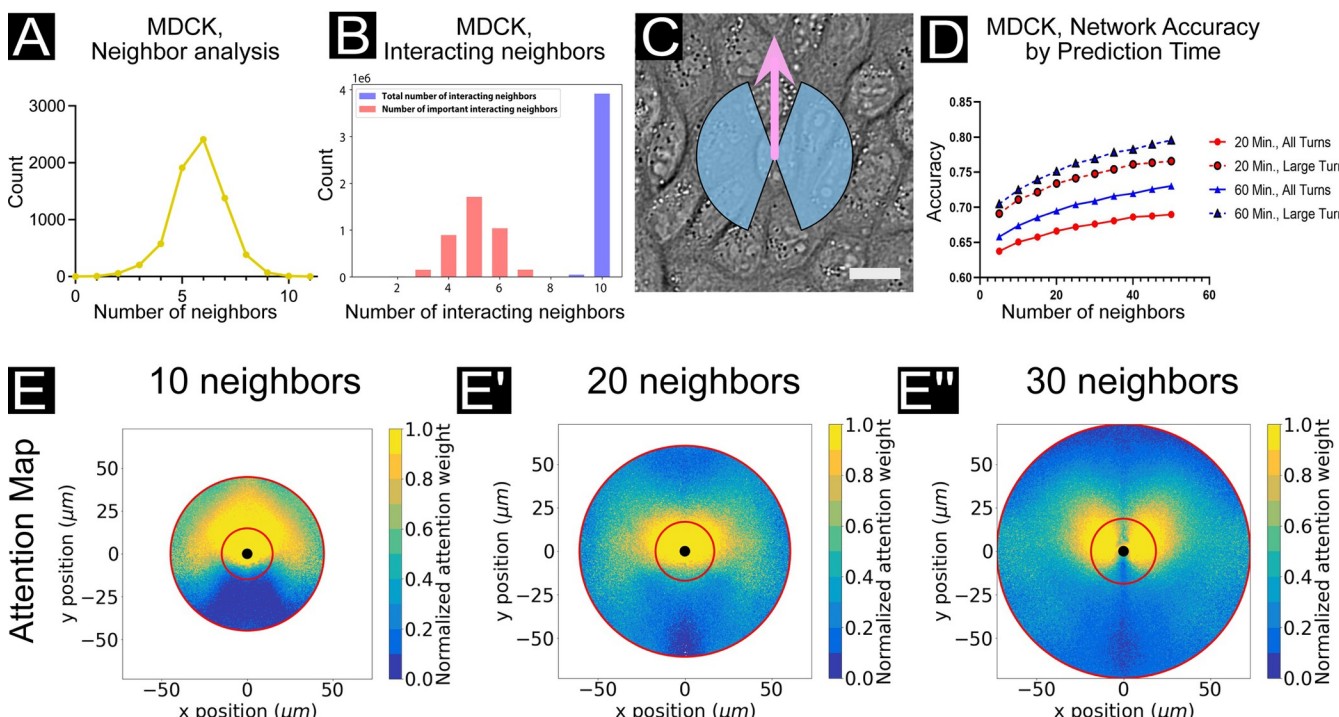

**Fig 3. Local vs. long-range interactions in MDCK epithelia (bulk regime).** (A) The number of nearest neighbors based on an analysis of 1165 cells using the ImageJ/FIJI [47] BioVoxxel plugin [48] (see *Methods*). A peak can be observed at 6 nearest neighbors. (B) Histograms of total interacting cells (blue) and "important" interacting cells (red), as determined by a function utilizing the network aggregation weights (*W*) to estimate the most influential neighbors to learned focal cell dynamics. (C) A snapshot of MDCK cells with blue region indicating the extent of "large" turns (±20–160°) according to the focal cell trajectory, as indicated by the pink arrow. Scale bar represents 20 µm. (D) Network accuracy plots as prediction time and number of input neighbors is varied. Solid lines reflect accuracy scores for all turning angles in the focal agent trajectory; dashed lines reflect only large turns (±20–160°, see C). Accuracy increases with both number of neighbors encompassed by the network and prediction time. Cell trajectory timesteps were fixed at 10 minutes. (E, E', E") Attention maps for networks encompassing 10 (*E*), 20 (*E'*), and 30 (*E"*) neighbors. Plots shown here are analogous to Fig 2C', with cell trajectory timestep of 10 minutes. As the number of neighbors taken into consideration by the network increases, a wider spatial range of interactions may be considered for forward motion prediction. With an increased range from which dynamic information can be directly captured from neighboring agents, we can observe shifts in learned relative influence of neighbors; for example, as longer-range neighbors provide richer information pertaining to dynamic shifts in the forward direction than immediate forward neighbors. **See S9 Fig for the matching study in HUVEC endothelial cells**.

all cases, we distinguish accuracy results across all turning motions of the focal cell ("all turns") from accuracy results restricted to turning motions ranging from ±20–160° ("large turns") (see Fig 3C). This compensates for edge cases where a cell may turn only very slightly off the forward axis. Overall, we notice three distinct trends relating to neighborhood size, turn magnitude, and temporal variables and discuss each aspect of Fig 3D in turn here.

With respect to prediction time steps, we observed a clear trend in both MDCK epithelia and HUVEC endothelia where the network accuracy improved with increasing time-steps, with data from either 20 min or 60 min forward predictions shown (red and blue lines in Fig 3D; see attention maps in S11 Fig). While modest (~5–7% for MDCK), we hypothesize that this trend reflects the relatively high persistence of confluent cells in epithelia and endothelia (S12D Fig). More specifically, predicting ahead over shorter time steps (e.g. 20 minutes) is more susceptible to fluctuations in the cellular dynamics and noise in the tracking data, while predicting over longer timesteps (e.g. 60 minutes) should act to temporally filter out these fluctuations and better emphasize the directed nature of cell migration in these cell types. Additionally, cells will undergo smaller displacements over short time steps, likely resulting in more ambiguous cases at the logit boundary (directly forward of the focal agent) where small spatial variations may produce a change in left vs. right turn classification.

To explore the importance of turning angles and the logit boundary, we compared accuracy data for 'all turns' versus that for 'large turns', as defined earlier and highlighted in Fig 3D. This comparison clearly showed improved accuracy for larger versus smaller turns. Again, this is due to smaller turns being closer to the logit boundary (0˚) and more difficult to predict. This finding was borne out across all experiments presented here. Further, the concept of turn magnitude can clarify the relationship between cell type and accuracy as certain cell types favor much smaller turns than others. To emphasize this, we plotted a radial histogram of focal turn angles in S12A–S12C Fig, where it is clear that HUVEC endothelial cells favor smaller turning angles (higher persistence) than MDCK epithelial cells (see S12D Fig for persistence plots). This explains why the network is more accurate at predicting MDCK vs. HUVEC behaviors, as HUVEC motion will lie closer to the logit boundary.

Overall, the number of neighbors assessed by the network was the most influential variable on network accuracy—as the network was structured to account for larger sets of nearest neighbors, the accuracy increased monotonically (Figs 3D and S9D). This trend was also true across all epithelial and endothelial datasets we considered, with varying strength. For instance, MDCK attention maps were more strongly affected by neighborhood size than HUVEC maps were (Fig 3D vs. S9D Fig). To more clearly capture this, we compared attention maps for three different neighborhood sizes (10, 20, and 30 nearest neighbors; NN) in Fig 3E–3E" for MDCK cells. Increasing the neighborhood size from 10NN to 30NN resulted in a shift from a forward cone of influence to more of an axially symmetric lobular structure. This shift is further emphasized by the associated scatter plots of closest nearest neighbors and highest weighted neighbors (S8A-A", S8B-B", S8C–S8C" Fig, respectively). Again, we emphasize that the neural network will have access to trajectory data for each one of the $n$ neighbors, whether or not the real focal agent does, and that long-range interactions (such as chemosignaling) can be captured as long as they occur within the timespan of the trajectory data. Users must be wary of any unique boundary phenomena (sustained tissue outgrowth and moving fronts), which may be captured within the analyzed timeframe and can influence the learned importance of long-range neighbors.

## Context of network accuracy for collective cell migration

The link between network accuracy and neighborhood size reflects an important and counterintuitive design consideration since the cells we analyzed here, unlike fish, only have direct, physical awareness of their true contiguous nearest neighbors. Hence, while the accuracy increases with increasing number of nearest neighbors accounted for by the network, as more information can be obtained over a wider spatial range, an individual cell has a more limited *biological* sensing regime. Thus, an increase in accuracy with increasing neighborhood size may not reflect biological realities of the system, and may instead result from the network learning more longer-range interactions. Given this, it may be helpful to configure attention networks to match the desired biological questions or constraints rather than exclusively pursuing accuracy.

Typically, the objective is to obtain as high an accuracy result as possible for a given task for most deep learning problems. Here, by contrast, the objective is more nuanced: first, we are not interested in specifically using the predicted turning logit, but rather contrive the dynamics prediction task specifically in order to recover collective rules from the trained network weights in the form of interpretable attention maps. That is, the network only has to be "good enough" to learn the essential collective dynamics. Second, certain systems may be more challenging to learn, such as the HUVECs which tend towards small turning angles.

To account for these two difficulties, we compare the standard network accuracies to accuracies derived from a network trained using shuffled trajectories: specifically, where social but

not asocial data is shuffled for each trajectory. A difference in accuracy values indicates that the network captures collective phenomena. For MDCKs, the standard training accuracy was 64.3% for all turns, 70.1% for large turns, compared to the shuffled training accuracy which was 59.1% for all turns, 62.5% for large turns. For HUVECs, the standard training accuracy was 58.0% for all turns, 58.5% for large turns, compared to the shuffled training accuracy which was 53.4% for all turns, 53.1% for large turns. While we consider this accuracy increase to indicate learned collective dynamics, we hope that our work will encourage the development of richer dynamic prediction tasks and metrics to this end.

In addition to network structure modifications, we also assessed the importance of (1) sampling rate (time intervals between data points), and (2) the choice of input variables. To explore sampling rate effects, we compared our prior networks trained on data captured at 10 min/frame to new networks trained from scratch on data sub-sampled at 20 or 30 min/frame (S13 and 14 Figs for MDCK and HUVECs, resp.) In these experiments, the accuracy increases as the time delay is increased, most likely due to the access of the network to longer total time intervals due to the use of the same number of historical time steps. Finally, we blind the network to focal tangential acceleration and neighbor accelerations (S15 Fig), that is, we exclude these parameters as input to the network. The accuracy results are not significantly impacted by the exclusion of acceleration parameters. When we consider network performance in a complex system like an epithelium, we see that no single modification—temporal variables, neighborhood size, turn binning—accounts for more than a 10% improvement in performance at best, while all network conditions outperformed a random guess and generally presented similar overall trends, or rulesets.

As a final note, we emphasize that it is crucial to consider context when comparing accuracy results. For data taken from the same cell types under the same experimental conditions, increased accuracy results can provide useful information about which input variables may strongly impact turning dynamics. However, accuracy comparisons may provide less insight across cell types, such as in the case of HUVEC endothelial cells which have narrower turn angle distributions than MDCK epithelial cells (see S12A–S12C Fig), or differences in prediction task, such as short- vs. long-time prediction intervals, which can modify which neighbors are likely to influence focal agent dynamics. While we did perform parameter sweeps over key variables such as forward prediction time and number of neighbors considered, it was necessary to establish baseline conditions to present our findings. For all standard epithelial and endothelial experiments, unless otherwise stated, 10 total nearest neighbors were accounted for by the network (i.e. 10 pairwise-interaction subnetworks, 10 aggregation subnetworks), the time between trajectory points was 10 minutes, the prediction time interval was 20 minutes, and no parameter blinding was performed. Further, we restricted our core analyses to these standards in order to best learn temporally local cell dynamic "decisions"—with 20 minutes corresponding to the approximate time it takes these cells types to move approximately half a nuclear-length within a confluent ensemble based on our data (Fig 1D)—and additionally to sufficiently encompass spatially local neighboring cells, as a function of classical neighbor analyses as in Fig 3A.

## Limiting cases: mesenchymal, metastatic cells lack coordinated collective rules

Our goal is to study collective behaviors in cells, so a natural question which arises is: how do these networks respond to cell types with apparently uncoordinated behavior? We explored this using metastatic breast cancer cells as a hallmark in many metastatic cancers is that cells undergo an epithelial-to-mesenchyme transition, effectively transitioning from more collective, epithelial cells to more individualistic mesenchymal cells [7]. We explored this here using

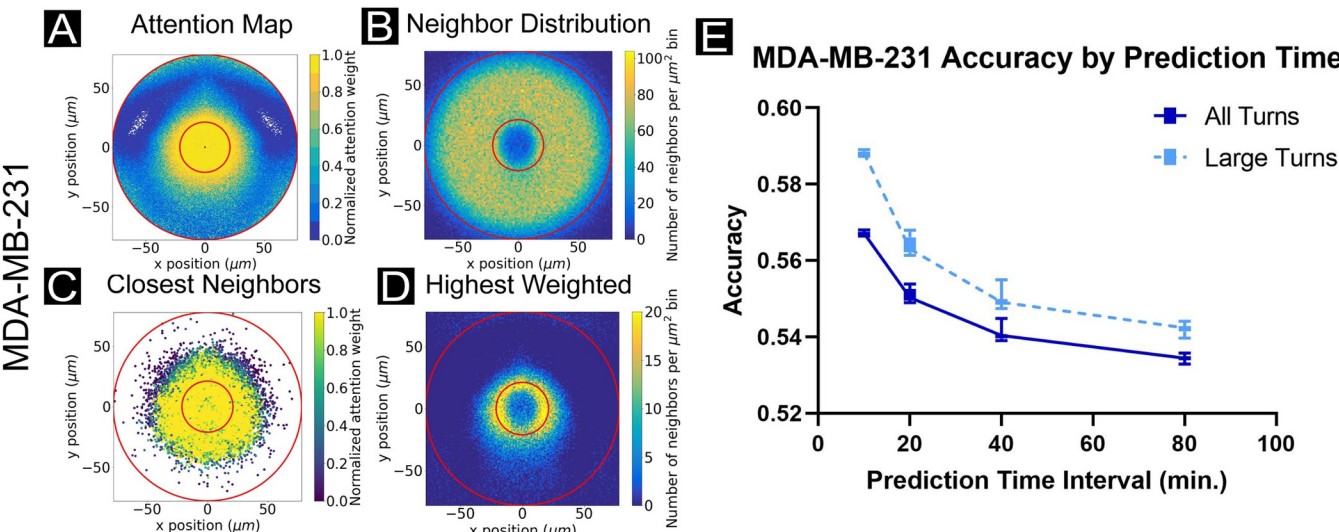

**Fig 4. Breaking coordination: attention in metastatic cancer cell line MDA-MB-231.** (A) Normalized attention weight contour plot, (B) neighbor location histogram, (C) closest neighbor scatter plot, as colored by normalized attention weights, and (D) histogram of highest weighted neighbors, with all plots analogous to those in Figs 2 and 3. Results shown for MDA-MD-231 cells with cell trajectory points taken every 5 minutes, and networks encompassing 10 neighbors with 20 minute prediction times. This cancer line functions as a control, as the cancer cells are highly uncoordinated, resulting in nearly equal attention weight applied to local neighbors in all directions. (E) Network accuracy plots as prediction time interval is varied, aggregated over networks accounting for 5–50 neighbors in increments of 5. Solid lines reflect accuracy scores for all turning angles in the focal agent trajectory; dashed lines reflect only large turns (±20–160˚). Accuracy decreases with increasing prediction interval and varies little as a function of neighbors observed by the network. Cell trajectory timesteps were fixed at 5 minutes.

the MDA-MB-231 cell line: a well-studied, highly aggressive triple-negative breast cancer (TNBC) cell type, which exhibits spindle-shaped morphology, and lacks strong cell-cell adhesion [49–51]. In contrast to the highly collective MDCK and HUVEC lines, the uncoordinated MDA-MB-231s function more like a negative biological control.

The attention plots and accuracy scores for the MDA-MB-231s are shown in Fig 4. The attention contour plot in Fig 4A highlights a radially symmetric influence regime around the focal agent, indicating that dynamics are more likely influenced by proximity alone (possibly a repulsion zone) than directed coordination. The histogram of neighbor locations (Fig 4B) confirms that the data are relatively consistently distributed about the focal cell, while the scatter plot of the closest neighbor locations, colored by normalized attention weights (Fig 4C) and histogram of highest weighted neighbors (Fig 4D) further emphasize the circular influence region lacking any more specific spatial signature. Here, the prediction time interval was 20 minutes, the time between trajectory points was 5 minutes, and 10 nearest neighbors in total were accounted for by the network structure.

As individual MDA-MB-231 cells lack cell-cell adhesion-mediated coordination, and exhibit low-persistence trajectories (S12D Fig), the ability of the network to predict future turning decreases with increasing prediction time interval (Fig 4E). The velocity autocorrelation (S12E Fig) plot drops off sharply within approximately 50 minutes, which is consistent with the drop-off in accuracy within the first approximately 50 minutes in accuracy vs. prediction time interval, as the system loses its dynamic 'memory' within this time interval. This accuracy drop-off is opposite the trend from more collective and persistent cell types where accuracy increases with increasing prediction time interval and is likely a hallmark of poorly coordinated cells. Additionally, accounting for larger numbers of nearest neighbors does not obviously impact the network accuracy results (S12F Fig). Again, since the agents are highly uncoordinated, the range of interacting cells does not affect predictive accuracy.

## Biophysical and biological variations affect the attention maps

Finally, we explore how collective cell migration rules vary across a large tissue and in different biophysical contexts. There is a growing appreciation in tissue biology that cells within a single tissue can exhibit different behaviors based on their locations within the tissue—supracellularity [2]. These differences can arise from local biological or biophysical properties, such as density-mediated jamming and contact inhibition of locomotion and proliferation [44,45]. Here, we explore these questions in two parts using our MDCK epithelial model. First, we examine the collective rules found in epithelial cells near either the outer boundary of a growing tissue or deep in the bulk of the tissue. Next, we look at how the rules change in response to maturation of the tissue and concomitant biophysical changes. Accuracy plots for the following data can be found in S16 Fig.

To characterize 'edge vs. bulk' dynamics, we defined analysis zones to demarcate cell trajectories in the bulk and edge regions, excluding those cell trajectories too close to the free boundaries to avoid biases caused by reduction in neighbors (see *Methods*). Independent deep attention networks were trained for each zone. The attention contour plot, closest neighbor location scatter plot, and highest weighted neighbor histogram from Fig 2 are shown again in Fig 5B–5D, and represent the dynamics in the bulk region. Neighbor location histograms are shown in S17 Fig. Fig 5B–5D' are the same visualizations for data from the edge region of the tissue. Structurally, the key difference in these attention maps is the relatively much higher importance of lateral neighbors for cells at the expanding edges of a tissue. The neighbor location histogram plots (see S17 Fig) confirm that this difference is *not* due to a *lack* of cells in front of the focal cell. Rather, we hypothesize that agents directly in front of the focal agent near the edge of the tissue tend to have less influence over the turning behavior because as edge cells expand outward, the forward agents are more likely to displace outward, leaving space for the focal agent to follow yet not substantially impacting turning decisions overall where lateral cell-cell adhesion likely mechanically influences cell behavior. In both cases, agents forward-and-to-the-sides impact focal cell turning behaviors, with little impact from rear neighbors. Noting that the edge regions contained ~30% fewer cells overall than the bulk, we also provide attention maps representing reduced training datasets (by including only a fixed number of trajectories) for the MDCK bulk region and edge region cases (as well as for the HUVEC cell system), allowing us to ensure a sufficient amount of data was collected (S18 Fig). The qualitative nature of the attention maps may or may not change with an increasing training set size; in general, users should assess whether or not the model itself adequately predicts the collective forward system dynamics for their use case.

Having varied cell context across the tissue, we then varied cell context with respect to time and crowding. As an epithelium matures, it undergoes multiple rounds of cell division that drive the bulk density higher until it reaches a critical point where cell division is inhibited and migration slows due to jamming and contact inhibition of proliferation and migration signaling [15,45], S4 Movie. To study this here, we compared attention behaviors for cells in the bulk of a relatively 'young' tissue to those of a more mature tissue. The four attention plots associated with the post-contact-inhibition case are shown in Fig 5B–5D" for comparison to the first row of plots (Fig 5B–5D) which are representative of tissues prior to contact inhibition. These attention contour plots of mature, dense epithelia (Fig 5B") demonstrate a much shorter range zone of influence, reflecting the increased packing and reduced motility for cells in these tissues. The neighbor location histogram (Fig 5C", red lines) also confirms the denser packing of the tissue: more nearest neighbors proportionally lie within a thin annulus near the focal agent. Finally, beyond simply reducing the interaction length, focal cells in high density tissues

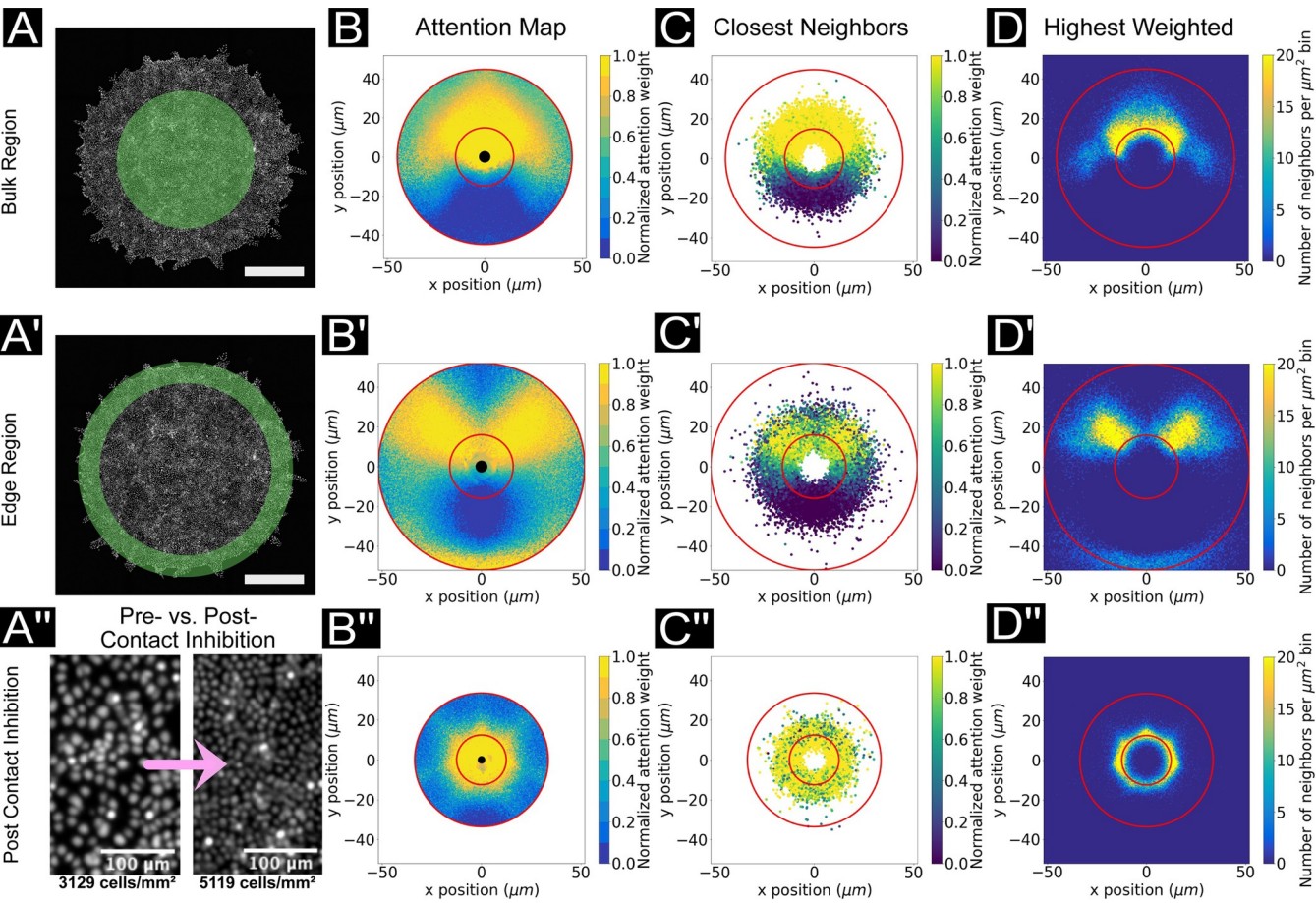

**Fig 5. Biophysical modifications and attention.** (A, A') For these experiments, cell trajectory data is extracted from either the bulk region (A) or the edge region (A') of the tissues. Scale bar represents 1 cm. (A") Representative nuclei images of tissues before and after contact inhibition. Scale bars represent 200 μm. (B*, C*, D*) Attention map, closest neighbors scatter plot and histogram of highest weighted points, as before. (B, C, D) Network trained on MDCK cell trajectories taken from a circular ROI in the center of an expanding tissue, prior to contact inhibition. Plots are representative of the bulk region (see *Methods*). (B', C', D') Network trained on MDCK cell trajectories taken from an annulus along the outer region of a circular expanding tissue, prior to contact inhibition. Plots are representative of the edge region of the tissue (see *Methods*). (B", C", D") Network trained on MDCK cells in the bulk region, after contact inhibition. Plots are representative of a jammed tissue (see *Methods*).

uniformly distribute their attention in all directions (Fig 5D"), in stark contrast to the biased attention patterns observed in the earlier, more motile state of the tissue.

Interestingly, these data raise an important point about comparison between, and analysis of, attention maps. For instance, the attention maps of highest weighted neighbors appear visually similar at first glance between metastatic (Fig 4D) and jammed epithelia (Fig 5D") despite vast differences in cell behaviors. However, quantifying these attention maps by radial averaging revealed a key difference (S19A Fig). Specifically, MDCK cells exhibited a strikingly localized radial zone of 'high attention' neighbors that, critically, does not overlap with the location of the focal cell. This makes sense and indicates a hard-core of repulsion around the focal cell. However, MDA-MB-231 metastatic cells exhibited a broad attention zone that overlapped with the focal cell, consistent with cells literally crawling across the focal cell and suggesting less structured motion overall. A comparison of MSD between dense epithelia and metastatic cells emphasized this lack of structure (S19B Fig). This was further supported by comparison of the accuracy plots (Figs 3D and 4E) that showed that MDCK prediction accuracy increased with time lags while MDA-MB-231 accuracy decreased with increasing time lags.

### Detection of collective behavior changes in response to external perturbations

Finally, we investigated the impact of modifications to cell signaling on the attention maps. Here, we perturbed the canonical MDCK model cell system with a drug selected to impact epidermal growth factor (EGF)—TAPI-1—which has been shown to inhibit spatial signaling and extracellular signal-regulated kinase (Erk) activation, and thereby collective migration [52,53]. The results of this experiment (see Methods) are shown in S20 Fig and indicate a striking difference relative to unperturbed tissues (e.g. Fig 2). Specifically, EGF disruption nearly abolished the relative importance of immediate forward neighbors, shifting the focus to immediate left and right neighbors. This shift in relative attention away from the forward neighbor and towards the lateral neighbors likely reflects the network detecting underlying biomechanical differences induced by EGFR/Erk signaling disruption as prior molecular studies have connected MDCK front-rear polarity to EGFR/Erk signaling [54]. While future work may be needed needed to verify and elucidate the specific molecular mechanisms, there are two key points to emphasize. First, this resulting shift in attention is not easily apparent from visual observation alone, emphasizing the importance of attention works for detecting subtle, collective responses to perturbations. Second, the attention network detected and clearly highlighted a connection between Erk and neighbor coordination without any foreknowledge of biased assumptions from the user, which makes it a powerful tool for hypothesis generation and screening of complex cellular dynamics datasets.

## Discussion

### Basic rules of collective cell attention can be learned from trajectory data

We demonstrated that deep attention networks can learn core rules of collective cell behaviors given only cellular trajectory data, offering a complementary approach to traditional biophysical and statistical methods for analyzing collective cell behaviors. In blood vessel endothelial cells (HUVEC), where strong leader-follower dynamics are visually observable, the attention maps emphasized the overwhelming learned relative influence of cells directly in front of the focal cell, rather than lateral or rearward neighboring cells. Again, these results do not follow from either classical correlation analyses or biological morphology and protein localization data. [40] In epithelial cells (MDCK), where cell-cell interactions are more complex and tend to result in large-scale correlated motion domains within the tissue, the relative influence region was much broader and encompassed neighboring cells forward and to the sides, with minimal influence from cells behind the focal agent. In more individual, metastatic breast cancer cells (MDA-MB-231), which are highly uncoordinated and function as a biological control, attention maps reflected a lack of learned influence in any particular direction in contrast to the collective HUVEC and MDCK cells, with influence confined to a small region in close proximity to the focal cell. Our visual attention map results, increased accuracy scores compared to networks trained on shuffled trajectories, and accuracy trends as a function of network modifications–such as increases in prediction time intervals—indicate that the deep attention networks are effectively recovering collective influence regions.

Broadly, attention analysis reflects the integrated effects of a variety of cell-cell coupling mechanics such as traction forces, cell-cell junctions, jamming, and chemical signaling [55–57]. While attention maps cannot deconvolve these effects, they can still highlight the resulting phenotypes. Extending the earlier discussion, the powerful forward neighbor influence in HUVEC attention maps derive mechanistically from the polarized VE-cadherin structures (Fig 2) that generate front/rear tension with no lateral coupling [40]. Similarly, the shift in attention maps with young versus old MDCK epithelia reflects the classic biophysical jamming transition, while the distinct influence pattern in attention maps taken at the growing edge of

epithelia likely reflect the unique traction force and monolayer stress states at epithelial boundaries. Attention mapping may eventually help to connect biophysical mechanisms to collective behavior 'rules', as is hinted at in the ability of the network to detect how chemical disruption of EGFR/Erk signaling reprograms collective attention (S20 Fig).

Overall, attention maps can add new context and build on classical correlative or ensemble approaches, allowing for improved interpretability of collective motion dynamics. Fundamentally, the success of the intuitive power of the attention maps is a function of the success of the deep neural network model to capture agent-agent relationships within the collective, from which the learned, relative influence of each neighbor is obtained. Therefore, we can think of the learned relationships between agents as "causal" in that the learned model reflects real-world system dynamics.

### Limitations of existing metrics and network design

Recall that our approach draws on tools originally developed for analyzing schooling fish, and so we note that translation to complex, orders-of-magnitude larger populations of interacting cells is not perfect. In particular, our work highlights the need for novel metrics and performance benchmarks to validate network success. We utilize the deep attention network structure to both capture rich dynamic relationships and expose meaningful attention weights for interpretation. Establishing more rigorous criteria to assess if meaningful collective behaviors are captured would be of great value towards transitioning similar techniques into standard practice, such as: (1) the development of a suite of biologically-grounded perceptual range targets for canonical cell types; (2) establishment of different learning goals beyond simple turning decisions; and (3) application of new network architectures and strategies such as reinforcement learning.

Deep attention network accuracies may be augmented by providing information about the system which is inaccessible to the biological agent, such as dynamic information about cells beyond the focal cell's physical sensing boundaries (Fig 3D), or the use of long-term historical data (S13 and S14 Figs). Moreover, we are applying a tool originally developed for the analysis of independent, physically separated agents (e.g. fish) with wide, non-contact based perceptual fields (vision and pressure wave detection) to a 2D confluent monolayer in which cells are physically contacting one another. Thus, network inputs, network structure, and metrics of success must be carefully designed to ensure the learned dynamics are reflective of the biological system.

### Concluding remarks

Here, we characterize the application of deep attention networks to the recovery of cell-cell influence within a collective setting. We apply the technique to data collected from well-studied epithelial cell lines with distinct collective behaviors and in distinct biophysical settings. We compare accuracy results as a function of different training, data sampling, and sensory range settings, and explore how different geometric and biological contexts can alter the underlying 'rules' and corresponding attention maps. We highlight the need for improved network structures and performance metrics; however, we are optimistic about the potential for deep attention networks and related machine learning methods to reveal collective rules beyond the capabilities of classical group analysis methods.

## Methods

### Ethics statement

Our study involved standard mammalian cell type the use of which is approved via Princeton IBC committee, Registration #1125–18. MDCK-II wild-type and Ecad:RFP cells were a gift

from the Nelson Laboratory at Stanford University. HUVEC cells expressing VE-cadherin were a gift from the Hayer Laboratory at McGill University. Wild-type HUVEC cells were purchased through Lonza. MDA-MB-231 human breast cancer cells were a gift from the Nelson Laboratory at Princeton University.

## Cell culture

MDCK-II cells were cultured in low glucose DMEM supplemented with 10% Fetal Bovine Serum (Atlanta Biological) and penicillin/streptomycin as done previously [15]. HUVEC endothelial cells were cultured using the Lonza endothelial bullet kit with EGM2 media according to the kit instructions. MDA-MB-231 human breast cancer cells were cultured in DMEM/F12 (1,1) media [58] (Thermo Fisher Scientific, Life Technologies, Item #11330–032) supplemented with 10% Fetal Bovine Serum (Atlanta Biological) and penicillin/streptomycin. All cell types in culture were maintained at 37°C and 5% $CO_2$ in humidified air.

## Tissue preparation

Tissue samples were grown in 3.5-cm glass-bottomed dishes coated with an appropriate ECM. To coat with ECM, we incubated dishes with 50 μg/mL in PBS of either collagen-IV (MDCK, MDA-MB-231; Sigma) or bovine fibronectin (HUVEC; Sigma) for 30 min 37°C before washing 3 times with DI water and air drying the dishes.

To pattern consistent circular tissues, ~3 μL of suspended cells were seeded into 9 mm$^2$ silicone microwells within each dish as described in [[44]] which allowed confluent monolayers to form. MDCK-II cells were seeded at a density of 1.8x10$^6$ cells/mL; HUVEC cells were seeded at a density of 0.8x10$^6$ cells/mL; and MDA-MB-231 cells were seeded at a density of 3.0x10$^6$ cells/mL. Then cells were allowed to adhere in the incubator (30 min for MDCK, 1 hr for HUVECs, 2 hrs for MDA-MB-231s), after which we added media and returned them to the incubator for 16 hrs prior to imaging. For contact inhibition samples, MDCK-II cells were seeded at a density of 4.2x10$^6$ cells/mL on 20mm$^2$ silicone microwells. After 30 min. incubation, tissues were continuously over 48 hrs to capture both pre-contact inhibition and post-contact inhibition state. For TAPI-1 experiments, MDCK-II cells were prepared as previously described, but 2 μL of TAPI-I (Selleck) at 10mM concentration in DMSO was added to each dish. For TAPI-1 validation experiment, MDCK FUCCI iRFP ERK-KTR cells were prepared with the same method without TAPI-1 treatment.

## Fluorescent imaging

We used the live nuclear dye NucBlue (ThermoFisher; a Hoechst 33342 derivative) with a 30 min incubation for nuclear labeling on standard MDCK, HUVEC, and MDA-MB-231 tissues and imaged with a DAPI filter set. For MDCK data collected for pre- and post-contact inhibition experiments, nuclear labels were reproduced using a convolutional neural network trained to reconstruct nuclei features from 4x phase contrast images of cells. Complete documentation including code and trained network weights for this tool may be referenced in [39]. Media was swapped and silicone microwell stencil was removed prior to imaging. Cadherin imaging was performed using conventional epifluorescence microscopy on a Nikon Ti2 equipped with a YFP filter set (HUVEC VE-Cadherin) and an RFP filter set (MDCK E-cadherin).

## Image acquisition

MDCK, HUVEC, and MDA-MB-231 data was collected on a Nikon Ti2 automated microscope equipped with either a 4X/0.15 phase contrast (HUVEC) objective or 10X/0.3 phase

contrast objective (MDCK, MDA-MB-231), and a Qi2 sCMOS camera (Nikon Instruments, 14-bit). An automated XY stage, a DAPI filter set, and a white LED (Lumencor SOLA2) allowed for multipoint phase contrast and fluorescent imaging. MDCK and HUVEC data were collected at 10 min/frame (49/140 frames in total, respectively), while MDA-MB-231 were given 5 min/frame (97 frames total), with temporal resolution increased for the MDA-MB-231 cells to improve tracking quality. Contact inhibition data were collected at 20 min/frame for 48 hours. The first 60 frames and last 60 frames are used as pre and post contact inhibition samples, respectively.

All imaging was performed at 37˚C with 5% $CO_2$ and humidity control. Exposures varied, but were tuned to balance histogram performance with phototoxic risk. Data with any visible sign of phototoxicity (blebbing, apoptosis, abnormal dynamics) were excluded entirely from training.

## Timelapse pre-processing and tracking

Timelapse movies of individual expanding tissues were processed using ImageJ/FIJI [47,59] prior to performing cell tracking via background subtraction and contrast enhancement. Tracking was performed using the TrackMate plugin in ImageJ [60], with "bulk" vs. "edge" tissue regimes initially differentiated using a circular ROI concentric with the tissue with radial extent 80% of the tissue radius. Cell trajectories were generated and shortened tracks were excluded to account for boundary effects: for instance, cells from the bulk tissue regime migrating into the edge regime. Trajectories were normalized, by translation to the trajectory arena center and scaling, and smoothed as in [[9]], with cell velocities and accelerations determined using finite differences. The bulk spatial regimes were further reduced by 20% prior to training, while the edge spatial regimes were reduced by 10% of the maximal tissue growth prior to training, again to mitigate edge effects. When trajectories were subsampled, cell trajectory positions were sliced to use every $n$th value in time; when tissues at different growth stages were analyzed; full trajectory datasets were sliced to include data spanning the required time ranges.

The protocol for determining nearest neighbors, velocities and accelerations, turning angles, and shuffled trajectories was identical to the protocol in [[9]]; however, the size of the training dataset was reduced in order to increase the size of the validation and test datasets (50%/30%/20% by timelapse splits). In total, 13 individual tissue timelapse movies were collected for the HUVEC cell system; 15 movies for each MDCK cell system, and 17 movies for the MDA-MB-231 cell system. Independent dishes were held out from the training dataset for testing purposes. With data pre-processing, each timelapse movie for the HUVEC system resulted in approximately 70,000 data points, compared to approximately 300,000 for MDCKs and approximately 100,000 for MDA-MB-231s.

## Network training and analysis

The attention network structure, logit probabilities, loss function, and training hyperparameters were identical to those described in [9], here again implemented using Keras with a TensorFlow backend [61,62], yet with a standard 1000 epochs per training cycle and early stopping. The structure of the deep attention network extends to include $n$ pairwise-interaction subnetworks and $n$ aggregation subnetworks, where $n$ is the number of nearest neighbors accounted for by the network. The standard value of $n$ is 10 unless otherwise specified. Each pairwise interaction block consists of a fully connected network with 3 layers of 128 neurons each followed by rectified linear unit (ReLU) operators, plus a final output layer of one neuron. These blocks are also anti-symmetrized. The weight function blocks are identical except that

there is an exponential function after the final one-neuron layer, and the input is accepted in a y-reflection-invariant form. The output of the weight blocks multiply the output of the corresponding pairwise interaction blocks for each neighboring agent. All pairwise interaction blocks share the same weights. Sample training loss plots are shown in S4 Fig. Training was performed on a desktop using an NVIDIA GeForce GTX 1070 Ti GPU or in a cluster environment with an NVIDIA Tesla P100 GPU. As in Francisco J. H. Heras et al. [9], the attention network logit was used to determine a logit indicating whether the focal agent will turn left or right after a fixed time interval. The network input consisted of asocial information, specifically the speed, $v$, tangential acceleration, $a_\parallel$ and normal acceleration, $a_\perp$; and social information pertaining to a set number of nearest neighbors to the focal agent, specifically relative position, $x_i$ and $y_i$, velocity, $v_{i,x}$ and $v_{i,y}$, and accelerations, $a_{i,x}$ and $a_{i,y}$. We performed experiments "blinding" the model to the focal tangential acceleration and neighbor accelerations (both normal and tangential), such that these variables would not be included as input to the model, yet no significant effect was observed on accuracy (see S15 Fig).

All plots were generated using Python unless otherwise indicated. The representative cell trajectories in Fig 1A–1C were generated using the TrackMate plugin ImageJ. The mean speeds, MSD and persistence plots in Fig 1D and 1E were generated using TrackMate trajectories, with persistence calculated as (displacement)/(traveling distance) and MSD calculated by MATLAB script (MSDAnalyzer). The cell position snapshot in Fig 1F plots a single random focal cell, indicated by a central ellipse, and relative positions in space of its neighbors as a function of nuclei centroids, colored by normalized attention weight output by the network according to their trajectory data. Neighboring cell direction is indicated by elongated axis of the ellipse, and nuclei centroids were used to generate Voronoi cells.

Attention maps (e.g. Fig 2A) were generated by selecting 10,000 random focal agents in the test set and interpolating the attention weights assigned to every neighbor of every focal agent to produce a contour plot. Attention weights are normalized in the range of 0–1 based on the maximum and minimum attention weight values in the test set; only relative weight strength is considered here. The radius of innermost black circle indicates the smallest radial distance from any focal agent to its closest neighbor. The thin red circles indicate the region in which the bulk of the neighboring points lie in space. The neighbor positions are converted into radial distance values to determine radii between which 5%-95% of the data falls; these radii are indicated via the thin red lines on both attention maps and neighbor distribution maps. The latter (e.g. Fig 2B) were generated using the same 10,000 focal agents and their neighbors and binning their $(x, y)$ coordinates to produce a 2D histogram. Closest neighbor location plots (e.g. Fig 2C) were produced by utilizing the same 10,000 focal agents yet sorting their neighbors by radial distance to the focal agent; only those closest neighbors were plotted in space, and points were colored by normalized attention weight. Highest weighted neighbor histograms (e.g. Fig 2D) were generated using the same 10,000 focal cells, yet only binning the $(x, y)$ coordinates for the neighbor with the highest weight for each focal cell. The focal turning angle radial histogram (S12 Fig) was generated using the same 10,000 focal cell trajectories and binning angles by 10˚.

Neighbor analyses were performed using the ImageJ BioVoxxel toolbox [48]. First, cell boundary binary images were obtained by processing nuclear fluorescence data using the 'Find Maxima' routine in ImageJ with 'segmented particle' output. Next, we used BioVoxxel neighbor analysis with the 'particle neighborhood' approach and a neighborhood radius of 2 pixels. Interacting neighbor plots (e.g. Fig 3B) were produced as described previously [9], with the important neighbors recovered as a function] of the inverse of the typical attention weight (Eq 2) as presented previously [63]. All accuracy results are reported on the complete test set.

### Collective simulation analysis

To validate if deep attention networks recover differences in attention in known cases, we trained them using simulated trajectories. This data was generated using a commonly used model for collective motion—the Vicsek model. The model was set up according to the original paper [64]. The parameters used are as follows: $\eta = 0.1$, $L = 50$, $N = 3000$, $r = 1$, $v = 0.3$, $t_{MAX} = 200$, $\delta t = 1$. For some simulation cases, changes were made to the model in order to reduce the perceptual zone of each agent. In the modified Vicsek model, a focal agent's heading will only be affected by other agents within its perceptual zone. We tested four cases defined by the agents' perceptual zones: full 360˚ perception, 60˚ perception in front of the agent, 120˚ perception in front of the agent, and 60˚ perception behind the agent. Each dataset contained 15 simulations in the training set and 3 in the test set. The networks were trained using 15 nearest neighbors and 1 prediction time step.

## Supporting information

**S1 Fig. MSD analyses.** Mean squared displacement (MSD) over time. (A) Linear-scale MSD to emphasize distinct differences in MSD trajectories; shaded zones indicate the weighted standard deviation of the individual MSD trajectories (see MSDAnalyzer software). (B) Log-scale of MSD for a more traditional rendering of the MSD that highlights the long-lag caged behavior of MDCKs.
(JPG)

**S2 Fig. Neighbor importance to learned turning dynamics, additional snapshots.** Individual agents are plotted in space ($x$, $y$) and colored according to relative attention weight ($W$) as in Eq 1 for HUVECs (left) and MDA-MB-231 cells (right). Cell position is representing using nuclei centroids and black lines indicate Voronoi cells (see Methods).
(JPG)

**S3 Fig.** Attention maps for collective simulation (Vicsek model) Individual attention maps were produced for agent trajectories generated via (A) the classical Vicsek model with full radial perception, and Vicsek models in which the perceptual range between collective agents is constrained to (B) 60˚ (±30˚) behind the focal agent, (C) 60˚ (±30˚) ahead of the focal agent, and (D) 120˚ (±60˚) ahead of the focal agent. The attention maps are able to capture these ranges directly from trajectory data alone. See *Methods*.
(JPG)

**S4 Fig. Representative loss functions from the attention network training process.** Early stopping was enabled, so that if the validation loss did not decrease within a set number of epochs, the training process was terminated. Validation loss was noisier when training the network on MDA-MB-231 data, in which there is reduced cell-cell coordination.
(JPG)

**S5 Fig. HUVEC and MDCK attention maps with increasing training epoch.** The attention maps for (A-C) the standard HUVEC cell system and (A'-C') the standard MDCK cell system are shown, as (left to right) the number of training epochs is increased from 10 epochs, to 100 epochs, and finally to the fully trained system. The test accuracy for the HUVEC system after 10 epochs is 57.2% (57.3% large turns); while for the HUVEC system after 100 epochs it is 58.2% (58.7% large turns). The test accuracy for the MDCK system after 10 epochs is 54.1% (57.3% large turns); while for the MDCK system after 100 epochs it is 53.9% (56.9% large turns).
(JPG)

**S6 Fig. HUVEC and MDCK attention maps with speed thresholding.** Attention maps are shown for the (A-C) HUVEC cell system and (A'-C') MDCK cell system. We compare the full attention map for each system (C, C') utilizing all available data points, to those data points where the focal agent speed is either (A, A') below or (B, B') above a threshold speed chosen to be the median speed value for all focal agents in the system. No meaningful structural difference was observed when speed thresholding was performed in this way.
(JPG)

**S7 Fig. Closest neighbor histogram plots for main cell systems.** The histogram representation of the closest neighbor plots for (A) HUVEC, (B) MDCK, and (C) MDA-MB-231 cell systems are shown, analogous to the closest neighbor scatter plots represented in Figs 2E, 2E', and 4C, respectively.
(JPG)

**S8 Fig. MDCK (bulk) neighbor distribution, closest neighbor, and highest weight maps.** Plots shown are analogous to the neighbor distribution, closest neighbor, and highest weight neighbor maps shown in Fig 2D–2F', yet corresponding to the 10, 20, and 30 neighbor networks with attention maps as in Fig 3D–3D".
(JPG)

**S9 Fig. Local vs. long-range interactions in HUVECs.** (A) The number of nearest neighbors based on an analysis of 1115 cells using the ImageJ/FIJI [47] BioVoxxel plugin[48] (see Methods). A peak can be observed at 3 nearest neighbors. (B) Histograms of total interacting cells (blue) and "important" interacting cells (red), as determined by a function utilizing the network aggregation weights (W) to estimate the most influential neighbors. (C) A snapshot of HUVEC cells with blue region indicating the extent of "large" turns (±20–160˚) according to the focal cell trajectory (indicated by the pink arrow). Scale bar represents 20 μm (D) Network accuracy plots as prediction time and number of input neighbors is varied. Solid lines reflect accuracy scores for all turning angles in the focal agent trajectory; dashed lines reflect only large turns (±20–160˚). Accuracy increases with both number of neighbors encompassed by the network and prediction time. Cell trajectory timesteps were fixed at 10 minutes. (E, E', E") Attention maps for networks encompassing 10 (left), 20 (middle), and 30 (right) neighbors. Plots shown here are analogous to plots shown in Fig 3, with cell trajectory timestep of 10 minutes. As the number of neighbors taken into consideration by the network increases, a wider spatial range of interactions may be considered for forward motion prediction.
(JPG)

**S10 Fig. Complete MDCK bulk region network accuracy plot.** Network accuracy plots as prediction time and number of input neighbors is varied. Solid lines reflect accuracy scores for all turning angles in the focal agent trajectory; dashed lines reflect only large turns (±20–160˚). Accuracy increases with both number of neighbors encompassed by the network and prediction time. Cell trajectory timesteps were fixed at 10 minutes.
(JPG)

**S11 Fig. MDCK (bulk) attention maps, 60-minute prediction time interval.** Representative attention weight contour plots are shown for MDCK cells with networks accounting for 10 neighbors in total (A) and 30 neighbors in total (30) with prediction time intervals of 60 minutes. For all conditions, normalized weight maps are shown and are analogous to the 20 minute prediction time interval attention maps shown in Fig 3D and 3D".
(JPG)

**S12 Fig. Focal cell turning angle distribution and persistence.** A radial histogram of turning angles from focal cell trajectories, shown for (A) HUVECs, (B) MDCK cells in the bulk region, and (C) MDCK cells in the edge region (from the same tissues; see *Methods*). HUVEC angles tend to fall closer to vertical (0˚). (D) Persistence plot for all main cell systems indicating "directedness" by orientation over time. The persistence plot here highlights the tendency of the HUVECs in particular to proceed in a single direction; shaded zone represents standard deviation (see *Methods*). (E) Representative velocity autocorrelation for MDA-MB-231 cell system as an additional measure of the lack of dynamic persistence (generated using MSDAnalyzer). (F) MDA-MB-231 network accuracy is largely independent both of neighbor number and of time steps.
(JPG)

**S13 Fig. Network accuracy plots with trajectory subsampling: MDCK.** Network accuracy is shown as a function of number of neighbors encompassed by the network and time delay between cell trajectory points. (A) displays accuracy for a prediction time of 40 minutes, with 10 (blue) and 20 (green) minute time delays, resulting from subsampling of the initial trajectory results. (B) displays accuracy for a prediction time of 60 minutes, with 10 (blue), 20 (green), and 30 (red) minute time delays. Solid lines reflect accuracy scores for all turning angles in the focal agent trajectory; dashed lines reflect only large turns (±20–160˚). Accuracy increases as time delay is increased; in this experiment, the same number of historical steps is utilized, so subsampled trajectories include data spanning longer total time intervals.
(JPG)

**S14 Fig. Network accuracy plots with trajectory subsampling: HUVEC.** Network accuracy is shown as a function of number of neighbors encompassed by the network and time delay between cell trajectory points. (A) displays accuracy for a prediction time of 40 minutes, with 10 (blue) and 20 (green) minute time delays, resulting from subsampling of the initial trajectory results. Solid lines reflect accuracy scores for all turning angles in the focal agent trajectory; dashed lines reflect only large turns (±20–160˚). Accuracy increases as time delay is increased; in this experiment, the same number of historical steps is utilized, so subsampled trajectories include data spanning longer total time intervals.
(JPG)

**S15 Fig. Network accuracy plots with input acceleration blinding.** Network accuracy is shown as a function of number of neighbors encompassed by the network, prediction time, and input parameters to the network. Either the standard inputs are utilized (lighter colors, see *Methods*), or the model was blind to focal tangential acceleration and neighbor accelerations (darker colors; i.e., these parameters were excluded from model inputs). (A) displays accuracy for MDCK cells, (B) for HUVECs. Solid lines reflect accuracy scores for all turning angles in the focal agent trajectory; dashed lines reflect only large turns (±20–160˚). Accuracy is not substantially changed as a function of acceleration blinding.
(JPG)

**S16 Fig. Accuracy results for MDCK cells, biophysical modifications.** (A) Network accuracy plots as prediction time and number of input neighbors is varied for both bulk (darker colors) and edge (lighter colors) regions within a confluent MDCK tissue. Solid lines reflect accuracy scores for all turning angles in the focal agent trajectory; dashed lines reflect only large turns (±20–160˚). Accuracy results tended to be slightly higher in the bulk region. (B) Network accuracy plots as prediction time and number of input neighbors is varied for the same MDCK tissues prior to (lighter colors) and after (darker colors) contact inhibition. Accuracy results were

higher prior to contact inhibition.
(JPG)

**S17 Fig. Neighbor distribution plots for MDCK biophysical variations.** Histograms showing the distribution of data points (neighbor cell locations) from which the attention maps in Fig 5B,B',B" were generated.
(JPG)

**S18 Fig. Training set reduction: attention maps.** The training set size was reduced by limiting the number of total trajectories for (A-C) the HUVEC cell system (10,000 / 100,000 / 433,063 trajectories respectively); (A'-C') the MDCK bulk region cell system (100,000 / 1,000,000 / 2,082,519 trajectories respectively); and (A"-C") the MDCK edge region cell system (100,000 / 1,000,000 / 1,451,150 trajectories respectively). Accuracy results for reduced training set cases were as follows: For HUVECs, accuracies were (A) 59.0% (59.4% large turns) and (B) 59.2% (59.2% large turns). For MDCK (bulk region), accuracies were (A') 66.7% (73.0% large turns) and (B') 67.8% (74.9% large turns). For MDCK (edge region), accuracies were (A") 66.1% (72.1% large turns) and (B") 65.3% (72.1% large turns).
(JPG)

**S19 Fig. Distinguishing and interpreting visually similar attention maps between metastatic and jammed epithelial cells.** (A) Radial distributions of the most important neighbors is plotted for jammed MDCK tissue and MDA-MB-231 tissue. The most important neighbors of jammed MDCK are focused on ~10–20 µm zone while MDA-MB-231 tissue has a much broader distribution of the most important neighbors that also covers the focal cell, indicative of cells crawling over each other and a lack of repulsion. (B) MSD comparison between MDA-MB-231 and highly dense, jammed MDCK cells indicating how the MSD can complement the attention maps to reveal underlying differences.
(JPG)

**S20 Fig. MDCK attention plots with cell signaling modifications via TAPI-1.** TAPI-1 was added to the standard MDCK cell system to inhibit cell-cell signaling (see *Methods*). (A-D) Plots shown are analogous to the attention map, neighbor distribution, closest neighbor, and highest weight neighbor maps shown in Fig 2C'–2F'. In comparison to the standard MDCK cell system, the attention maps reveal the loss of the relative influence of forward neighbors to the focal agent; however, "lobing" (relative influence of forward left/right agents) remains. The test accuracy was 68.5% for all turns, and 76.2% for large turns. (E-F) Representative images of MDCK cells immediately before and 2 hours after treatment with TAPI-1, respectively. Cells show lower ERK activity (higher nucleus intensity) after treating TAPI-1.
(JPG)

**S1 Movie. HUVEC, MDCK, and MDA-MB-231 representative data.** S1 Movie shows a phase-contrast timelapse of HUVEC cells, imaged at 4x magnification, with fluorescent stained nuclei overlaid. S2 Movie shows a phase-contrast timelapse of MDCK cells, imaged at 10x magnification, with fluorescent stained nuclei overlaid. S3 Movie shows a differential interference contrast (DIC) timelapse of MDA-MB-231 cells, imaged at 10x magnification, with fluorescent stained nuclei overlaid.
(AVI)

**S2 Movie. HUVEC, MDCK, and MDA-MB-231 representative data.** S1 Movie shows a phase-contrast timelapse of HUVEC cells, imaged at 4x magnification, with fluorescent stained nuclei overlaid. S2 Movie shows a phase-contrast timelapse of MDCK cells, imaged at 10x magnification, with fluorescent stained nuclei overlaid. S3 Movie shows a differential

interference contast (DIC) timelapse of MDA-MB-231 cells, imaged at 10x magnification, with fluorescent stained nuclei overlaid.
(AVI)

**S3 Movie. HUVEC, MDCK, and MDA-MB-231 representative data.** S1 Movie shows a phase-contrast timelapse of HUVEC cells, imaged at 4x magnification, with fluorescent stained nuclei overlaid. S2 Movie shows a phase-contrast timelapse of MDCK cells, imaged at 10x magnification, with fluorescent stained nuclei overlaid. S3 Movie shows a differential interference contrast (DIC) timelapse of MDA-MB-231 cells, imaged at 10x magnification, with fluorescent stained nuclei overlaid.
(AVI)

**S4 Movie. MDCK post-contact-inhibition representative data.** S4 Movie shows MDCK tissue after contact inhibition, imaged at 4x magnification, with overlaid nuclei predictions produced using a neural network (see *Methods*). This movie is from the dataset as S2 Movie, but it shows the complete progression from an early confluent tissue to a late stage, mature tissue with full contact inhibition and jammed cells.
(AVI)

## Acknowledgments

We appreciate the advice in preparing this manuscript from Drs. Polavieja and Heras at the Champalimaud Foundation.

## Author Contributions

**Conceptualization:** Julienne LaChance, Daniel J. Cohen.

**Data curation:** Julienne LaChance, Jens Clausen.

**Formal analysis:** Julienne LaChance, Kevin Suh, Jens Clausen, Daniel J. Cohen.

**Funding acquisition:** Daniel J. Cohen.

**Investigation:** Julienne LaChance, Kevin Suh, Daniel J. Cohen.

**Methodology:** Julienne LaChance, Kevin Suh, Daniel J. Cohen.

**Project administration:** Daniel J. Cohen.

**Resources:** Daniel J. Cohen.

**Software:** Jens Clausen.

**Supervision:** Daniel J. Cohen.

**Validation:** Julienne LaChance, Kevin Suh, Jens Clausen, Daniel J. Cohen.

**Visualization:** Julienne LaChance, Kevin Suh, Jens Clausen, Daniel J. Cohen.

**Writing – original draft:** Julienne LaChance, Daniel J. Cohen.

**Writing – review & editing:** Julienne LaChance, Kevin Suh, Daniel J. Cohen.

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
