## [Decision Letter · Decision Letter 0]

8 Oct 2021

Dear Cohen,

Thank you very much for submitting your manuscript "Learning the rules of collective cell migration using deep attention networks" for consideration at PLOS Computational Biology.

As with all papers reviewed by the journal, your manuscript was reviewed by members of the editorial board and by several independent reviewers. In light of the reviews (below this email), we would like to invite the resubmission of a significantly-revised version that takes into account the reviewers' comments. With your revision, please provide a point by point response to the reviewers' concerns, paying particular attention to 1) contextualizing the approach a it more thoughtfully, perhaps by expanding the physics vs machine learning discussion to include a more general discussion of issues in the study of micro to macro like what might be gained by inferring rules from data beyond your point about higher dimension relationships, 2) better discussion of biophysical limitations of the approach 4) clarifying some confusion around the kinds of cell interactions that were considered and 5) what the migration rules imply about what cells must "know" to utilize them. 

We cannot make any decision about publication until we have seen the revised manuscript and your response to the reviewers' comments. Your revised manuscript is also likely to be sent to reviewers for further evaluation.

Sincerely,

Jessica C. Flack

Associate Editor

PLOS Computational Biology

Feilim Mac Gabhann

Editor-in-Chief

PLOS Computational Biology

Reviewer's Responses to Questions

**Comments to the Authors:**

Reviewer #1: This is a particulaly interesting article at the intersection of machine learning and collective phenomena in biology. In my opinion, it certainly meets the bar for interest, depth, and creativity for publication in PLoS Computational Biology.

Congratulations to the authors on a very interesting and compelling piece of work. In my opinion, this article is of wide interest to a number of communities, and it's an example of precisely what PLoS Comp Bio can be publishing.

I have some questions and concerns that I would like the authors to address; these are all in the discussion section. I have an optional suggestion which would require more work; if the authors choose not to do this optional suggestion, I'd ask them to add a paragraph talking about the fact that this optional suggestion is a great idea and should be done by someone.

A note: the article is largely descriptive. Rather than test a particular hypothesis about the nature of cell navigation, their goal is to explore the different ways in which cells alter their trajectories in response to their social context. This is OK!

This is also a tool-confirmation paper. It is good that the cancer cells are messed up, because cancer is messed up. And I appreciate the ways in which the authors tie their descriptive findings to other knowledge in the literature. However, the finding that endothelial cells are NN while epithelial are long-range seems to be something we already knew (or should have been able to guess) -- it's the intuitively right answer, not an informative one. That's OK!

****** First, I have two requests for revision. These are just additions to the discussion.

1. to what extent are these attention networks capturing *causality*, rather than simple correlation? It is nice, for example, that these networks detect that the forward direction matters more than the rear (despite the existence of correlations to the rear). But what do we know about how these attention networks capture the causal effects? Are they just capturing the "more important" correlations, or is there something new in play? Are the attention networks using time, for example, as a proxy for causality (not a bad heuristic, of course!)

I don't expect the authors to have an answer here, but my suspicion is that this is just a more refined version of earlier correlative studies. Some discussion and analysis of exactly what's going on, even just in an exploratory mode, would be good in the discussion.

2. it's not clear to me what the detection of long-range interactions means. As the authors note, the only direct physical coupling is to nearest neighbours. So what does it mean when the system is finding that long range matters more?

It seems there are (at least) two possibilities.

(A) there's chemosignalling; the chemical signals attenuate with distance, but there are more cells at longer range, and the second effect dominates. If this is true, then it seems like we're getting real biological information out of this process -- we're learning about the different signalling mechanisms.

(B) we're picking up epiphenomenal correlations; the nearest neighbour is setting the agenda, but it's doing so for both directions. The focal cell is correlated with the NN, the NN is correlated with things forward, and so it looks like the focal cell is receiving instructions from longer distances.

Talking through these possibilities would give the reader a better sense for the biological insights that might emerge.

****** And, I have one suggestion for the authors to investigate: what happens with simulated data? Let's say that you create a fake tissue, in silico, with cells that are following nearest neighbour rules (or have some other attention kernel).

Can you recover that kernel? How well can you do so? What do you get right about the kernel? For example, can you get accidental long-range attention windows even when the underlying dynamics are nearest neighbour? I don't think this needs to be exceptionally long.

A simple in silico experiment with two different navigation rules (one NN, one more long-range, perhaps roughly matching the HUVEC and MDCK cases), and the results of the attention network method, would be enough.

I think this could really help make the paper more widely compelling for readers, in part because most people are not going to go through the trouble of reproducing the analysis.

I don't want to say this is a mandatory edit, because I think it is a lot of work. However, if the authors choose *not* to do this, then they need to add a paragraph talking about how they *didn't* do this analysis, why it's a good idea, etc. -- just punting to future work, but making it clear that it's a question in play.

We want there to be some match between inference and reality, and a simulation is a nice way to check that things aren't going off the rails. Without that check, questions remain about the ways in which things could go wrong (e.g., remark #2 above.)

Reviewer #2: In this manuscript, LaChance et al. present an approach to learn aspects of the interactions between cells based on deep attention networks directly from experimentally measured cell trajectories. Specifically, trajectories of monolayers of different cell types (HUVEC, MDCK, MDA-MB-231) are passed through a deep attention network to identify the relative importance of neighbouring cells (quantified by their weight in the network) to predict the turning behavior of any given cell in the monolayer. Here, the authors largely follow the methodology of ref 9 in the paper. With this approach, the authors infer "attention maps" which give the average importance of neighbours as a function of polar angle from the considered cell. Interestingly, the different cell types have different attention maps: HUVEC cells have much more focused maps, indicating larger importance of head-neighbours. In contrast, MDA-MB-231, which are known to have a more random migration phenotype, have completely isotropic attention maps. While this approach is new to cell migration research and interesting, these findings are not interpreted or put into context with existing cell migration literature.

Overall, this study presents a new way to analyse cell migration based on trajectory data, which could present an important tool in the future. However, the method is not tested convincingly on benchmark data, where the interactions are known, and it is unclear what new insights are gained from applying it. Most importantly, the attention maps are inferred, but then never interpreted in depth. Thus several questions remain unanswered: What new things have we learned about cell migration with this analysis? Why should others apply this method in the future? What is the advantage of this method over existing analysis and inference methods?

Despite these shortcomings, I believe that with revisions, the paper could be suitable for publication. Specifically:

Major

- The paper lacks a discussion of the biophysical implications of the findings. An interesting aspect of the attention maps is that many collective cell migration models based on active particles assume a radial symmetry of the interactions: cells interact with forward neighbours just as much as with backward or lateral neighbours (see e.g. 10.1371/journal.pcbi.1002944 ; 10.1073/pnas.1219937110 and many other papers that implement alignment interactions inspired by the Viscek model). Such interactions can also be inferred directly from cell trajectory data, without relying on machine learning (see https://doi.org/10.1073/pnas.2016602118). The findings in this study seemingly contradict this assumption - however it is not clear whether they really contradict it, or whether this is an artefact of the method. Would an attention map for a Viscek-like model for cell migration in the flocking regime that correctly identify the radial symmetry of the interactions? In this parameter regime, the model should capture the cell data at the level of MSD, velocity cross correlations, and order parameter, but it would still be based on radially symmetric interactions. But, the attention maps should still correctly infer the radial symmetry. Then, using a model where these interaction in fact depend on the angle from the cell, the attention maps should also infer this angle correctly. Only if this is the case can the attention maps really be used to learn something new about cell migration. If it is not the case, then it is unclear how to interpret attention maps.

- In the introduction there is a lot of discussion about velocity cross correlations between cells, yet this quantity is never presented for the data. Could the authors show the velocity cross correlations for the 3 cell types in Fig. 1? Both the mean speed and the MSD are really single cell statistics and don't quantify collective motion.

- the language around attention is used very loosely in the paper (e.g. line 59, 104), and often seems to suggest that the cells really "pay attention to their neighbours". There is no notion of this in the literature and so it would have to be defined carefully. The wording should be much more careful to not mix up animal systems like fish with unconscious systems like cells.

- similarly, the paper would benefit from a discussion of how the concept of attention can be interpreted in the context of cell migration: how does it connect to the concepts usually invoked in the literature like active particle interactions, traction forces, monolayer stresses, ...

- lines 102-104: the authors argue against ensemble analyses, but then proceed to generat ensemble averaged attention maps. What do they mean with ensemble analyses? The arguments that follows is not convincing: why are ensemble averages not informative about interactions?

- Fig 4E: isn't this a trivial result for trying to predict a persistent random walk with a deterministic model? How does the prediction time interval compare to the persistence time of the cells (calculated e.g. from MSD or velocity autocorrelation)?

- the abstract is very confusing: there is a wealth of literature on expressing cell migration rules in interpretable form (e.g. everything on Contact Inhibition of Locomotion, alignment interactions etc.). The concept of a focal cell is not explained. Attention is not defined, and must be as it's a new concept for cell migration research.

Minor

- In line 4-5 the historical treatment seems to contradict the dates on the cited papers: were velocity correlations really first used on animal data and then "repurposed" for cells?

- In Fig 1E, the massive green area is not labeled or explained. The MSD should in addition be plotted on a loglog scale, to back up the claims in lines 9-11.

- line 132 should say experimental model systems rather than models to avoid confusion

- line 205 figure reference seems to be mixed up

- line 313 what is the logit boundary? is it defined anywhere?

- the concept of a focal cell could be better introduced (it's just the cell that's being focused on, it's not a special cell within the monolayer like a leader cell)

Reviewer #3: This work focuses on applying a deep attention network developed by Heras, et al. to coordinated cellular migration to compare behaviors across cell types. The parameters controlling coordinated cellular migration are challenging to intuit from existing data, and deep learning offers an unbiased approach with no assumptions to developing new metrics to interpret data about multicellular migration. The manuscript is very clearly written and transparent about the advances and current limitations. However, we have several concerns to be addressed before publication.

Major Concerns:

Innovation, originality, and importance to the field are publication criteria for PLoS Computational Biology. This manuscript could be strengthened by a novel finding instead of applying an existing method that largely confirms what is known. This could go to the extent of examining multicellular migration in an understudied cell system. However, addressing the other major concerns could also address this concern, and so we do not want to prescribe a specific way of addressing this concern.

How extracellular signaling impacts the model assumptions is unclear. The work appears to assume that physically-linked cells are the only important ones for attention maps, but cells such as MDCK cells are known to release EGF during collective migration. Perturbing extracellular signaling with flow and testing the model's response could be an exciting test of the limits of this approach.

The results would be improved by making a clearer distinction throughout between predictive power in the model and causative biological influence. For example, in the discussion of accuracy vs biological relevance, higher accuracy means that you have better predictive power if you know info about cells farther away from you. This is not the same as "this cell knows what's happening 3 cell lengths away from it". As the manuscript does mention higher accuracy does not always yield rules with biological relevance, it would be helpful to present (in supplements) weight maps generated from the models in Figure 2C and C’ as training accuracy goes up. As accuracy goes up during training, do the patterns in attention maps become more clear, or do differences emerge during the training process?

The manuscript would benefit from further exploration of the models, specifically the pi function, which should show how neighbors influence the focal cell, and the weight parameters other than xi,yi (speed, for instance). To address this, we have the following suggestions:

Include a supplementary exploration of pi as in Heras, et al. Figure 2 to show if the influences of cells in different regions differ among cell types or just the weights.

This could be used, similar to Heras, et al., to assign regions of parameter space as attraction/repulsion/alignment zones, if they exist. This would test the authors' proposal (line 422) that the MDA-MB-231 high weights represent a repulsion zone.

This could also help address an important question: the weight maps look highly similar for the bulk MDCK cells (described as 'trapped') and the MDA-MB cells (described as 'random'). Do the network outputs allow us to distinguish these two cases in any way?

Similarly, the attention maps are focused on strength of influence vs xi, yi, but W is a function of speed and acceleration as well; somewhere it would be useful to show these effects. Do some cell types pay more attention to faster cells, and others to nearer cells? What about the other inputs?

In the Methods, although we recognize that much of the detail can be found in Heras, et al. 2019, the authors should provide additional key information in the text. In particular, it would be better to define the network structure of the pairwise interaction function (how many layers, how many nodes on each layer, fully connected or not, etc.). Also, it would be more helpful if the authors provide more detailed information about data: exactly how many videos are used, roughly how many trajectories are in each video and of what duration, are data divided into training, validation and test groups by experiment, by tissues within the experiment, by trajectory, or by parts of trajectories?

Figures 2 and 4 would benefit from clarification about how the attention maps and closest neighbors are normalized. From the figures, they appear to be normalized to the maximum values in the heat maps. If that is the case, it would be nice to additionally show the absolute value of these heatmaps and to see whether there are differences between different cell types.

It is not clear from the Methods, but it would be interesting to see the model trained on one experiment and tested on a separate experiment with the same cell type. Or, attention maps from replicate networks trained independently on separate experimental replicates. This may be a substantial amount of work, but is important to the claim that the network is learning cell-type-specific behaviors. If this was done, this should be clarified in the Methods.

An additional analysis that could improve the manuscript is to test the dependence of the results on the number of cells and duration of trajectories used. For example, how many cells are needed to train such a network and how does accuracy vary w/ # of training data points? This is particularly relevant to the potential use of the model distinguishing different parts of the tissue (e.g. bulk vs edge) - how many edge cells do we need to be confident in the comparison? How will the results be affected if we have more bulk than edge cells?

Minor Concerns:

The font size on figures is too small on axis numbers and labels, as well as legends.

Why is the qualitative form of the influence map so different when the # of neighbors varies (e.g. Fig 3E*)?

Line 410 suggests the goal of using the MDA-MB-231 cells was to test if even in apparently uncoordinated cells there is an "underlying behavioral mode"; by the end of the paragraph these cells are designated a "negative biological control". This is a subtle point, but which is it?

The paragraph starting at line 359 could better emphasize that the shuffling method shuffles social but not asocial data for each trajectory. This is important because it helps explain that the small increases in accuracy reflect social data alone.

A histogram format for the closest neighbors plots would help with comparison to the highest weighted plots.

Typos:

Line 135: importance  important

Line 184: additional “.”

Line 256, 257: Fig S3 and Fig S5

Line 433: (C) closest neighbor scatter plot

Methods: line 591 says 3 uL and line 595 says 4 uL but they appear to be for the same experiment

Line 644: “[9]”

Line 688: “]\\”

Line 650-651: Reference not in the reference format (Heras et al.)

Line 660: "TrackMate plugin *in* ImageJ"

**Have the authors made all data and (if applicable) computational code underlying the findings in their manuscript fully available?**

Reviewer #1: Yes

Reviewer #2: Yes

Reviewer #3: Yes

PLOS authors have the option to publish the peer review history of their article (what does this mean?). If published, this will include your full peer review and any attached files.

Reviewer #1: No

Reviewer #2: No

Reviewer #3: No
---

## [Decision Letter · Decision Letter 1]

6 Mar 2022

Dear Cohen,

Thank you very much for submitting your manuscript "Learning the rules of collective cell migration using deep attention networks" for consideration at PLOS Computational Biology. As with all papers reviewed by the journal, your manuscript was reviewed by members of the editorial board and by several independent reviewers. The reviewers appreciated the attention to an important topic. Based on the reviews, we are likely to accept this manuscript for publication, providing that you modify the manuscript according to the review recommendations. These recommendations are relatively minor but please attend to them and note them in your return cover letter. 

Sincerely,

Jessica C. Flack

Associate Editor

PLOS Computational Biology

Feilim Mac Gabhann

Editor-in-Chief

PLOS Computational Biology

[LINK]

Reviewer's Responses to Questions

**Comments to the Authors:**

Reviewer #2: The authors have addressed all my comments, including the control study on the Viscek model, which has improved the paper. I therefore recommend publication. I would like to congratulate the authors on a great manuscript that I am confident will be of interest to a broad audience.

Reviewer #3: We appreciate the authors have put in a considerable amount of effort to improve the clarity and scientific rigor of their manuscript, and think some of the new results are very exciting. We recommend publication after the following minor points are addressed.

1. The results concerning the impact of extracellular signaling on model assumptions are very exciting (Figure S20). However, these findings really demonstrate the power of the model and are potentially under-discussed in the text. For example, the finding that the focus shifts to left and right neighbors from the forward neighbors could be placed in more of a biological context even if the mechanism is unclear.

2. The section “Biophysical and biological variations affect the attention maps” may be more accurately renamed “Biophysical, biochemical, and biological variations affect the attention maps” with the addition of the results described in Figure S20.

3. We appreciate the authors for clarifying how the attention weights are normalized. It would be helpful to understand why they only consider relative strength important and if the absolutely strengths from different cell lines are comparable or not.

4. The new Figure S18 clearly illustrates in some cell types there are significant changes in the qualitative nature of the attention maps. It would be helpful if the authors expanded on why in some cases varying the trajectory number has such a large impact.

5. Line 512-513 should cite the results demonstrating that accounting for larger numbers of nearest neighbors does not obviously impact the network accuracy results.

6. The Figure 3 legend says to see Figure S6 for a matching study in HUVEC cells, but the text and supplemental figures suggest this should be referencing Figure S9.

**Have the authors made all data and (if applicable) computational code underlying the findings in their manuscript fully available?**

Reviewer #2: Yes

Reviewer #3: None

PLOS authors have the option to publish the peer review history of their article (what does this mean?). If published, this will include your full peer review and any attached files.

Reviewer #2: No

Reviewer #3: No

Figure Files:

Data Requirements:

Reproducibility:

References:

---

## [Editor Report · Decision Letter 2]

23 Mar 2022

Dear Cohen,

We are pleased to inform you that your manuscript 'Learning the rules of collective cell migration using deep attention networks' has been provisionally accepted for publication in PLOS Computational Biology.

Best regards,

Jessica C. Flack

Associate Editor

PLOS Computational Biology

Feilim Mac Gabhann

Editor-in-Chief

PLOS Computational Biology

---

## [Editor Report · Acceptance letter]

22 Apr 2022

PCOMPBIOL-D-21-01237R2 

Learning the rules of collective cell migration using deep attention networks

Dear Dr Cohen,

I am pleased to inform you that your manuscript has been formally accepted for publication in PLOS Computational Biology. Your manuscript is now with our production department and you will be notified of the publication date in due course.

With kind regards,

Livia Horvath
